# From Directions to Regions:
# Decomposing Activations in Language Models via Local Geometry

**Or Shafran** [1]   **Shaked Ronen** [1]   **Omri Fahn** [1]   **Shauli Ravfogel** [2]   **Atticus Geiger** [3]   **Mor Geva** [1]

## Abstract

Activation decomposition methods in language models are tightly coupled to geometric assumptions on how concepts are realized in activation space. Existing approaches search for individual global directions, implicitly assuming linear separability, which overlooks concepts with nonlinear or multi-dimensional structure. In this work, we leverage Mixture of Factor Analyzers (MFA) as a scalable, unsupervised alternative that models the activation space as a collection of Gaussian regions with their *local* covariance structure. MFA decomposes activations into two compositional geometric objects: the region's centroid in activation space, and the local variation from the centroid. We train large-scale MFAs for Llama-3.1-8B and Gemma-2-2B, and show they capture complex, nonlinear structures in activation space. Moreover, evaluations on localization and steering benchmarks show that MFA outperforms unsupervised baselines, is competitive with supervised localization methods, and often achieves stronger steering performance than sparse autoencoders. Together, our findings position local geometry, expressed through subspaces, as a promising unit of analysis for scalable concept discovery and model control, accounting for complex structures that isolated directions fail to capture.

## 1. Introduction

Disentangling the representations of language models (LMs) into causal interpretable units has been a hallmark of interpretability research (Sharkey et al., 2025; Geiger et al., 2024; Mueller et al., 2024). A growing consensus in re-

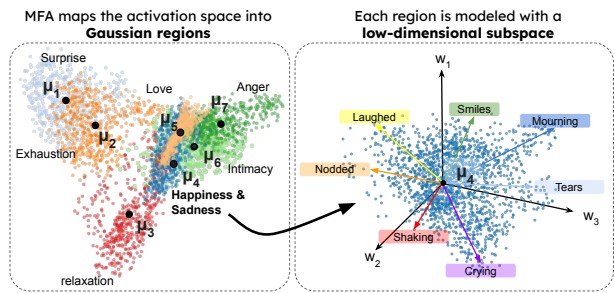

*Figure 1.* **MFA decomposes each activation into a region assignment and a within-region offset.** Left: the region structure is modeled by Gaussian components (centroids $\mu_k$), with complex concepts typically spanning multiple Gaussians – here, the broader *Emotions* neighborhood is spanned by several interpretable Gaussians. Right: each component is equipped with a low-dimensional subspace that parameterizes structured within-region variation.

cent years suggests global directions in activation space as candidate units, with many concepts empirically exhibiting linear structure (Ravfogel et al., 2020; Elhage et al., 2021; Gurnee et al., 2023; Nanda et al., 2023; Park et al., 2024). Consequently, much attention has been given to developing methods that disentangle activations into combinations of global directions (Yun et al., 2021; Bricken et al., 2023; Cunningham et al., 2023; Gao et al., 2025, *inter alia*).

However, such decomposition methods are tightly coupled to strong geometric assumptions (Hindupur et al., 2025) that overlook growing evidence of representations with more complex geometrical structures (Cai et al., 2021; Chang et al., 2022; Engels et al., 2025; Park et al., 2025; Gurnee et al., 2026). Specifically, nonlinear or multi-dimensional concepts are dispersed across many global directions with no built-in structure relating them to one another, so recovering the concept requires post hoc assumptions about which directions form a single representation (Chanin et al., 2025; Engels et al., 2025).

This limitation has recently driven a shift toward analysis units naturally modeled as subspaces rather than as isolated global directions (Sun et al., 2025; Huang & Hahn, 2025; Tiblias et al., 2025). Yet, while recent work shows meaningful geometric structure in activation space, how to turn these insights into practical tools for decomposition and steering remains an open challenge.

[1]Blavatnik School of Computer Science and AI, Tel Aviv University, Israel [2]New York University, New York, NY, USA [3]Goodfire. Correspondence to: Or Shafran <ordavids1@mail.tau.ac.il>.

*Proceedings of the 43rd International Conference on Machine Learning*, Seoul, South Korea. PMLR 306, 2026. Copyright 2026 by the author(s).

In this work, we tackle this challenge through a local-geometry lens, building on evidence that LMs exhibit local low-dimensional structure (Cai et al., 2021; Lee et al., 2025; Saglam et al., 2025). We propose a *scalable, unsupervised* method that partitions the activation space into regions and, within each region, learns a low-rank subspace that captures dominant modes of variation. Then, a given activation is decomposed into two compositional geometric objects: a region in activation space and a within-region offset (see Figure 1). To formalize the decomposition, we use classical statistical methods and employ Mixtures of Factor Analyzers (MFA; Ghahramani et al., 1996), a generative model that represents each region as a low-rank Gaussian distribution.

We apply our approach to Llama-3.1-8B (Grattafiori et al., 2024) and Gemma-2-2B (Team et al., 2024), training MFAs with 1K, 8K, and 32K components. Analyzing the discovered regions reveals two classes: *narrow* Gaussians that concentrate on a constrained lexical pattern (e.g., the word *"in"* in varying contexts), and *broad* Gaussians that encompass wide thematic topics (e.g., movies or emotions). Broad Gaussians often exhibit semantic local variation, while narrow Gaussians show more syntactic variance. Yet, with a larger number of components, Gaussians become narrower and their local variance differentiates based on context. Moreover, we observe that neighboring components tend to encode related semantics and, collectively, tile broader conceptual neighborhoods. These observations suggest that concepts may be realized not by a single component, but by constellations of nearby Gaussians that jointly cover a semantic, complex region.

Next, we contrast the decomposition induced by MFA with that of sparse autoencoders (SAEs), the predominant dictionary learning method. We find that MFA yields a simple decomposition in which both the assigned region and the local variation are highly interpretable. In contrast, SAEs rely on a single dictionary of global directions. In our experiments we found that on average 75% of the active features were not directly interpretable from the context.

Finally, we evaluate MFA's decomposition as a practical tool, showing it outperforms existing disentanglement methods on localization and steering benchmarks. For localization, MFA outperforms large-scale SAEs and various supervised baselines, beating Desiderata-Based Masking (strong supervised baseline) (De Cao et al., 2020; 2022; Csordás et al., 2021; Davies et al., 2023; Chaudhary & Geiger, 2024) on 5 out of 8 tasks across models, and often being competitive with the state-of-the-art DAS (Geiger et al., 2023). On steering, utilizing MFA centroids steers better than SAE features in the majority of settings, typically exhibiting a twofold gain on coherence and conceptual alignment. Together, these results indicate that MFA's mixture structure supports both causal localization and controllable generation.

To conclude, we propose a local-geometry view of activation space, partitioning it into low-dimensional regions and modeling the intrinsic modes of variation within each region. This approach yields an interpretable decomposition and scales gracefully to thousands of subspaces. Empirically, MFA outperforms existing unsupervised and supervised baselines on localization and causal mediation benchmarks, positioning local subspace structure as a promising unit of analysis for understanding how LMs organize information. We release our code and trained MFAs at: `https://github.com/ordavid-s/decomposing-activations-local-geometry`.

## 2. Preliminaries and Notation

**Factor Analysis (FA)** FA is a statistical method that models observed data with a small number of latent factors that explain correlations between variables. Unlike standard PCA[1], FA is a generative probabilistic model, which defines a likelihood for the data and explicitly models noise. Intuitively, the model assumes that most correlations among observed dimensions arise from a few underlying factors, while the remaining variation is dimension-specific independent noise. This yields a low-rank approximation that captures shared structure without requiring a full-covariance model. Formally, we assume each observed sample $\mathbf{x} \in \mathbb{R}^d$ is generated from the following generative model:

$$\mathbf{x} = W\mathbf{z} + \boldsymbol{\epsilon}, \tag{1}$$

with latent factors $\mathbf{z} \sim \mathcal{N}(0, I)$ and noise term $\boldsymbol{\epsilon} \sim \mathcal{N}(0, \Psi)$ with a diagonal matrix $\Psi$. The covariance of $\mathbf{x}$ is therefore $C = WW^\top + \Psi$, combining shared variation (via W) and independent noise (via $\Psi$).

Notably, $W$ is invariant to orthogonal rotations. For any $Q$ defining an orthogonal rotation, $WQ$ and W induce an equivalent Covariance $C$ since $(WQ)(WQ)^\top = WW^\top$. Thus, FA identifies the low-rank subspace $\text{span}(W)$, while the interpretation of individual axes depends on an additional rotation convention.

**Mixtures of Factor Analyzers (MFA)** MFA (Ghahramani et al., 1996) extends FA by allowing different regions of the representation space to express their own local geometry. Rather than a single FA modeling directions of global variation, MFA models the space as a collection of local low-dimensional Factor Analyzers[2]. This property is useful when the data exhibits factors of variation unique to different regions in the observation space, such as those

---

[1]While standard PCA is not generative, probabilistic PCA provides a closely related generative formulation, differing from MFA primarily in its noise model.

[2]MFA is a low-rank variant of GMMs, making it more efficient and providing a local low-dimensional structure.

observed in the activation space of LMs (Cai et al., 2021; Lee et al., 2025; Saglam et al., 2025).

To represent this, MFA introduces a discrete latent variable $\omega \in \{1, \ldots, K\}$ that indicates which FA component generated a sample. Each component $k$ models a different region of the representation space by having its own mean $\boldsymbol{\mu}_k$, which sets the center of the region. After centering by $\boldsymbol{\mu}_k$, variability within that region is described by an FA model with a component-specific matrix $W_k$, which determines the orientation of the component's local low-dimensional subspace. The columns of $W_k$, commonly referred to as the *loadings*, describe how latent factors translate into changes in the local region of the observation space: each column corresponds to one factor, and its entries specify how much each observed dimension changes when that factor varies.

The generative model is the same as for FA, with the addition of a component specific mean which anchors the FA to a region of the observation space. Conditioned on $\omega = k$,

$$\mathbf{x} = \boldsymbol{\mu}_k + W_k \mathbf{z}_k + \boldsymbol{\epsilon}, \tag{2}$$

which yields a component covariance:

$$C_k = W_k W_k^\top + \Psi. \tag{3}$$

Given all components, the overall density is

$$p(\mathbf{x}) = \sum_k \pi_k \, \mathcal{N}(\mathbf{x} \mid \boldsymbol{\mu}_k, C_k), \tag{4}$$

where $\pi_k$ is the *mixture weight* of component k. Each Gaussian therefore contributes according to how well its mean and subspace geometry explain the sample. See Ghahramani et al. (1996) for additional details.

## 3. Mapping the Activation Space with Mixtures of Factor Analyzers

We show how MFA can map regions of the activation space into a set of reusable and interpretable geometric units. These units reflect how the model organizes information in its latent space. Our approach is motivated by previous work (Coenen et al., 2019a; Cai et al., 2021; Lee et al., 2025; Saglam et al., 2025) showing that activations do not cover the entire activation space uniformly, but rather cluster semantically, where within-cluster variation is well approximated by a small number of degrees of freedom.

Therefore, we seek a model that (i) partitions the activation space into coherent regions and (ii) captures the intrinsic low-dimensional directions of variation within each region. MFA satisfies both of these desiderata: it achieves (i) by learning a mixture over components and assigning each activation to components via posterior responsibilities, effectively carving the activation space into regions. Moreover, it

attains (ii) as each component is a factor analysis model: a Gaussian whose covariance is parameterized by a low-rank subspace, so variation within that region is modeled along a small set of learned directions.

**Initialization** Given a set of activations $X \subset \mathbb{R}^d$ extracted from the residual stream at a fixed layer, we initialize an MFA with $K$ components and latent rank $R$ for each component. For simplicity, we use a uniform rank across components. This choice acts as a conservative approximation to the local intrinsic dimension of each region, capturing the dominant modes of variation while mitigating ill-conditioned loadings. We initialize the component means $\{\boldsymbol{\mu}_k\}_{k=1}^K$ by running $K$-means on $X$ and setting each $\boldsymbol{\mu}_k$ to the corresponding cluster centroid. The mixture weights are initialized uniformly, $\pi_k = 1/K$ for all $k$. For each component, we initialize the factor loadings $W_k \in \mathbb{R}^{d \times R}$ with random values sampled from $\mathcal{N}(0, 1)$, and set the (component-shared) diagonal noise covariance to $\Psi = I_D$. We also experimented with other initializations. See additional discussion in §A.

**Training** Each mixture component $k$ defines a Gaussian density over activations,

$$p(\mathbf{x} \mid k) = \mathcal{N}(\mathbf{x} \mid \boldsymbol{\mu}_k, C_k), \tag{5}$$

with the same covariance as Eq. 3. The mixture weights $\{\pi_k\}_{k=1}^K$ combine these component densities into the marginal likelihood

$$p(\mathbf{x}) = \sum_{k=1}^K \pi_k \, p(\mathbf{x} \mid k). \tag{6}$$

We learn the parameters $\theta = \{\boldsymbol{\mu}_k, W_k, \Psi, \boldsymbol{\pi}\}$ by minimizing the negative log-likelihood with gradient descent:

$$\mathcal{L}(\theta) = -\frac{1}{B} \sum_{i=1}^B \log \Big( \sum_{k=1}^K \pi_k \, \mathcal{N}(\mathbf{x}_i \mid \boldsymbol{\mu}_k, C_k) \Big), \tag{7}$$

where $B$ is the batch size. This objective allows us both to learn the clustering of the data and the local directions of variation together under one optimization problem.

**Component Assignment** To assign a given activation $\mathbf{x} \in \mathbb{R}^d$ to its best fitting component, we inspect the likelihood of component $k$ given the activation, normalized across all components. We denote this term as the activation's *responsibilities* where the responsibility of component k for the activation $\mathbf{x}$ is computed using Bayes theorem as,

$$R_k(\mathbf{x}) = p(k \mid \mathbf{x}) = \frac{\pi_k \, \mathcal{N}(\mathbf{x} \mid \boldsymbol{\mu}_k, C_k)}{\sum_i \pi_i \, \mathcal{N}(\mathbf{x} \mid \boldsymbol{\mu}_i, C_i)} \tag{8}$$

These responsibilities assign each activation to the component whose local subspace best explains it, allowing us to express the activation as a mixture of the components.

**Decomposing an Activation**   Each activation can be expressed using a dictionary of all the component means $\boldsymbol{\mu}_k$ and loadings $W_k$. Specifically, for a given activation $\mathbf{x} \in \mathbb{R}^d$, we compute the component's latent coordinates using the posterior mean under FA (Ghahramani et al., 1996):

$$\hat{\mathbf{z}}_k = Z_k(\mathbf{x} - \boldsymbol{\mu}_k) \qquad (9)$$

$$Z_k := \left(I_R + W_k^\top \Psi^{-1} W_k\right)^{-1} W_k^\top \Psi^{-1} \qquad (10)$$

Under Eq. 2, $\hat{\mathbf{z}}_k$ is the posterior-mean latent vector whose projection $W_k \hat{\mathbf{z}}_k$ best explains the residual $\mathbf{x} - \boldsymbol{\mu}_k$. Given the generative assumption of FA (Eq. 1), the activation is represented by a scalar weight $R_k(\mathbf{x})$ for each mean $\boldsymbol{\mu}_k$, and coordinates $\mathbf{z}_k$ for each axis $\mathbf{w}_{k,j}$ (the $j$-th column of $W_k$). Collecting these into matrices, the reconstruction can be written as a linear product:

$$\mathbf{x} \approx A \, \mathbf{b}(x) \qquad (11)$$

$$A := \begin{bmatrix} \boldsymbol{\mu}_1 \mid W_1 \mid \cdots \mid \boldsymbol{\mu}_K \mid W_K \end{bmatrix} \qquad (12)$$

$$\mathbf{b}(\mathbf{x}) := \begin{bmatrix} R_1(\mathbf{x}) \\ R_1(\mathbf{x})\,\hat{\mathbf{z}}_1(\mathbf{x}) \\ \vdots \\ R_K(\mathbf{x}) \\ R_K(\mathbf{x})\,\hat{\mathbf{z}}_K(\mathbf{x}) \end{bmatrix} \qquad (13)$$

Where $A \in \mathbb{R}^{d \times K(1+R)}$ is formed by concatenating *along the column dimension*; each $\boldsymbol{\mu}_k \in \mathbb{R}^d$ contributes one column and each $W_k \in \mathbb{R}^{d \times R}$ contributes $R$ columns. In contrast, $\mathbf{b}(\mathbf{x}) \in \mathbb{R}^{K(1+R)}$ is formed by concatenating *along the row dimension*; for each $k$, we append the scalar $R_k(\mathbf{x}) \in \mathbb{R}$ followed by the length $R$ vector $R_k(\mathbf{x})\,\hat{\mathbf{z}}_k(\mathbf{x}) \in \mathbb{R}^R$. Thus, each $\mathbf{x}$ is decomposed into activation coefficients of the shared dictionary of means and axes, and the entire reconstruction is a single matrix multiplication between the responsibilities and the components.

**Global versus Local Decomposition**   Most activation decomposition methods treat the residual stream as governed by a single set of global directions. Eq. 11 instead induces a region-conditioned parameterization. Each activation is described by *where it sits* in activation space, via responsibilities over centroids ($R_k(\mathbf{x})\,\boldsymbol{\mu}_k$), and *how it varies locally*, via within-component coordinates ($R_k(\mathbf{x})\,\hat{\mathbf{z}}_k(\mathbf{x})$).

Since the factorization within a component is rotationally invariant (§2), the meaningful object is not any single loading vector, but the local subspace $\mathrm{span}(W_k)$. This motivates a shift in the *unit of analysis*, moving from isolated global directions to local regions with their own low-rank geometry.

In the following sections, we use MFA to map the activation space of modern LMs. We train large-scale MFAs on residual-stream activations from Llama-3.1-8B (Grattafiori et al., 2024) and Gemma-2-2B (Team et al., 2024) (§4) and

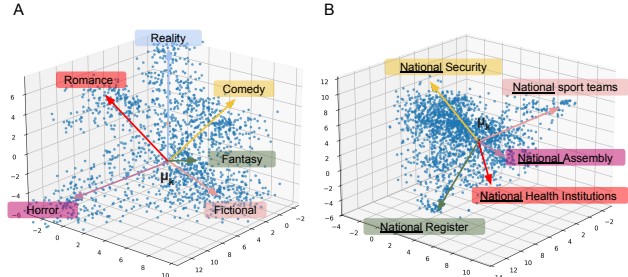

*Figure 2.* Example MFA Gaussians in the activation space of Llama-3.1-8B, visualized in 3D using three loadings as axes. (Left) A broad region spanning multiple movie genres, where the loadings separate genre-related themes. (Right) A narrow region centered on the token *National*, where the loadings capture context-dependent usage.

characterize the resulting regions and within-region variation. We then show that globally nonlinear concepts are often expressed as neighborhoods of multiple nearby Gaussians, and contrast MFA's decomposition with the global dictionary decomposition of SAEs (§5). Finally, we evaluate MFA on causal localization and steering benchmarks, where it outperforms unsupervised baselines and remains competitive with supervised methods (§6).

**Steering with MFA**   Let $\mathbf{x} \in \mathbb{R}^d$ be a hidden state from a layer at the last position of an input sequence. Fix an MFA component with centroid $\boldsymbol{\mu} \in \mathbb{R}^d$ and loadings $W \in \mathbb{R}^{d \times R}$, and let $\mathbf{v} \in \mathbb{R}^R$ be latent coordinates in the Gaussian's local subspace. We define the following interventions:

$$f_{\boldsymbol{\mu}}(\mathbf{x}) = (1 - \alpha)\mathbf{x} + \alpha\boldsymbol{\mu} \qquad (14)$$

$$f_{\mathbf{w}}(\mathbf{x}) = \mathbf{x} + \mathbf{W}\mathbf{v} \qquad (15)$$

Here $\alpha \in [0, 1]$ controls how strongly we move toward the centroid and $\mathbf{v}$ controls the direction and magnitude of the offset from the centroid. We interpolate toward $\boldsymbol{\mu}$ because it is an *absolute location* in activation space. In contrast, the loadings parameterize *within-region displacements* (directions in the centered space around $\boldsymbol{\mu}$), so we apply them additively as an offset.

## 4. Activation Structures Discovered by MFA

We train 12 MFAs on residual-stream activations from Gemma-2-2B (Team et al., 2024) and Llama-3.1-8B (Grattafiori et al., 2024). Specifically, we use layers 6 and 18 in Gemma, and layers 8 and 22 in Llama (approximately $1/3$ and $2/3$ of each model's depth), while varying the MFA scale with $K \in \{1K, 8K, 32K\}$ and fixing $R = 10$ as a practical tradeoff between capturing dominant local variation and keeping training tractable at a large scale. Each MFA was trained on 100M activations from The Pile (Gao et al.,

2020), initialized with K-Means on a random sample of 4M activations. For additional discussion on parameter choice, see §A. Analyzing the trained MFAs reveals rich structures in the activation space, with semantically coherent nonlinear manifolds captured as a collection of diverse, locally linear regions.

**Discovered Regions and Within-Region Variation**  We observe substantial diversity across components. Some regions are *narrow*, concentrating probability mass on a highly specific set of tokens or contexts, while others are *broad*, spanning a more comprehensive theme (Figure 2). The learned local subspaces also differ in the types of variation they capture. Mirroring observations from previous work (Coenen et al., 2019a; Simon et al., 2024; Park et al., 2025), within-region variation often reflects both *semantic* and *syntactic* differences. Some directions separate meaning and high-level content, while others track form and local structure, such as letter case and punctuation.

To quantify the types of structures captured by MFA, we sample 250 Gaussians from every MFA (600 in total), and annotate them as broad or narrow and their loadings as semantic or syntactic. Labeling of a Gaussian is done based on the theme of 25 sampled contexts with high likelihood under the Gaussian (Eq. 5). To label loadings, we first compute each context's coordinates within the Gaussian's subspace (Eq. 9). For loading $i$, we collect the 12 contexts with the largest value of the $i$-th latent coordinate $z_i$, and separately the 12 contexts with the smallest value. Since the sign of a loading is arbitrary, we label the two extremes separately, as we find both ends to be interpretable. We obtain the labels using an automated pipeline based on GPT-5-mini (Singh et al., 2025), which was validated against labels by NLP graduate students using statistical testing (Calderon et al., 2025). Although a "no pattern" option was provided as part of the annotation task, it was rarely selected by either humans or the LLM. Thus, we omit it from the results. Statistical testing results, annotation instructions and model prompts are provided in §B and §G respectively.

Figure 3 presents the annotation results, showing the ratios of broad/narrow Gaussians and semantic/syntactic loadings stratified based on the Gaussian type. In both models, larger $K$ increases the portion of narrow Gaussians, but the magnitude of this shift is model-dependent. In Gemma-2-2B, larger $K$ shifts mass toward narrow Gaussians, whereas in Llama-3.1-8B the partition remains predominantly broad even at $K$=32,000. This suggests that different models induce different notions of similarity in their activation spaces; Gemma's activation space tends to cluster primarily by token type (narrow), whereas Llama's clusters are driven more by semantics (broad). These differences may partly reflect architectural choices. Gemma-2-2B and Llama-3.1-8B differ in several respects, including normalization and atten-

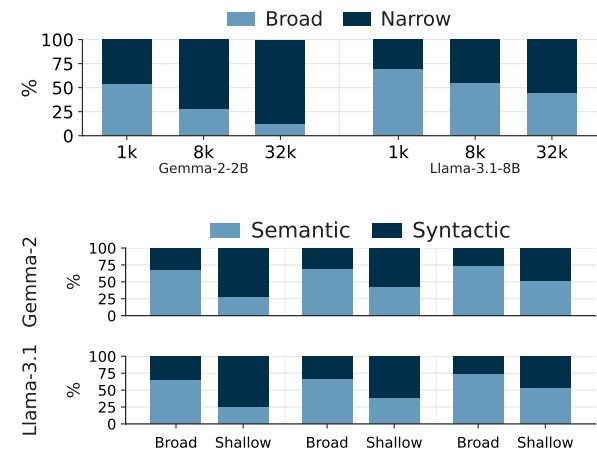

*Figure 3.* **Characterizing MFA regions.** (a) Broad vs. narrow regions differ across model families (Gemma skews narrow/token-driven; Llama stays mostly broad/semantic). (b) Semantic vs. syntactic loadings, split by broad/narrow components, become more semantic as $K$ increases, indicating more context-dependent within-region variation.

tion design, and such differences could affect the geometry of the residual stream. Additionally, we find that increasing $K$ not only raises the fraction of narrow regions, but also increases the frequency of semantic loadings, suggesting that within-component variation becomes more context-dependent. Across all settings, narrow Gaussians skew more syntactic, while broad Gaussians skew more semantic.

**Multi-Gaussian Concepts**  Consistent with prior work (Coenen et al., 2019b; Wiedemann et al., 2019; Park et al., 2025) showing that transformer representation spaces exhibit semantic organization, MFA components form coherent *semantic neighborhoods*, where nearby Gaussians tend to correspond to related meanings. Moreover, globally nonlinear concepts are often expressed not by a single component but by a *cluster* of neighboring Gaussians. MFA enables extracting these structures at scale. By treating components as nodes in a neighborhood graph, we construct a $k$NN graph using Euclidean distance between centroids and traverse local neighborhoods via BFS from selected components. Figure 1 illustrates one such neighborhood. Although individual components specialize in a narrower topic, such as happiness or surprise, together they form a unified "emotions" theme.

To quantify this property, we analyze whether nearby components tend to share semantic themes. For each component, we identify its three nearest neighboring components by Euclidean distance between centroids, gather high-likelihood contexts for the component and its neighbors, and prompt an LLM judge (gpt-5-mini) how many neighbors share the same theme. Across 670 components from our trained 8K

*Table 1.* **Centroid vs. loading token promotion.** Representative top-promoted tokens under centroid ($\boldsymbol{\mu}$) vs. loading ($\mathbf{w}_j$) interventions. In the *narrow* region, the centroid promotes a general "National" theme, while loadings separate subthemes. In the *broad* region, the centroid promotes genres, and loadings refine it into subgenres/media-specific patterns.

| Term | Top promoted tokens |
|------|---------------------|
| **Genres Gaussian** (Figure 2A, *broad*) | |
| $\boldsymbol{\mu}$ | `thriller, horror, sitcom, fiction, romance, fantasy, comedy, novel` |
| $\mathbf{w}_1$ | `fantasy, tale, RPG, adventure, realms` |
| $\mathbf{w}_2$ | `sitcom, television, TV, series, show` |
| $\mathbf{w}_3$ | `detective, espionage, spy, thriller, saga` |
| **"National" Gaussian** (Figure 2B, *narrow*) | |
| $\boldsymbol{\mu}$ | `Association, Institute, Museum, Newspaper, Infantry, Championship, Organization` |
| $\mathbf{w}_1$ | `Register, register, Historic, historic` |
| $\mathbf{w}_2$ | `League, Football, Hockey, football, hockey` |
| $\mathbf{w}_3$ | `Commission, Committee, Congress, caucus` |

MFAs, 61% had at least one same-theme neighbor, 36% had at least two, and 20% had all three. This supports the view that semantic concepts often extend across multiple nearby Gaussians, rather than being isolated in a single component. Concrete examples of such neighborhoods are in §F.1.

## 5. MFA vs. Dictionary Learning

Both MFA and dictionary learning are *generative models* that decompose representations into components. Ideally, such a decomposition should be "simple"—that is, each example should be explained by only a few interpretable components. To clarify how MFA's decomposition differs from dictionary learning, we conduct a side-by-side comparison with state-of-the-art SAEs.

**Global versus Local Decomposition**  We sample activations from Gemma-2-2B and Llama-3.1-8B for Wikipedia inputs, and compare their decompositions by our 8K MFAs (§4) and by the Gemmascope/Llamascope SAEs (Lieberum et al., 2024; He et al., 2024). We decompose each activation $\mathbf{x} \in \mathbb{R}^d$ with MFA into its responsibilities vector $\mathbf{b}(\mathbf{x})$ and the corresponding MFA components $A$ such that $\mathbf{x} \approx A\mathbf{b}(\mathbf{x})$ (Eq. 11). Similarly, for an SAE with hidden dimension $N$ we encode $\mathbf{x}$ into its SAE activations, $\mathbf{a} \in \mathbb{R}^N$ and the corresponding SAE features $F \in \mathbb{R}^{N \times d}$ such that $\mathbf{x} \approx aF$. For each method, we collect components with nonzero activation or responsibility and plot the cumulative reconstruction path along the top three PCA directions, sorting vectors by magnitude. This visualization shows how each method incrementally assembles its reconstruction.

Figure 4 visualizes the reconstruction by MFA (purple-blue) and SAEs (orange-yellow) for two representative examples. SAEs often use many features to reach the target, producing longer trajectories in PCA space. This reflects the dictionary learning geometry, SAEs represent an activation as a sparse sum of global directions, so the reconstruction is assembled via many incremental additions. In contrast, MFA trajectories consist of two segments. The first explains $\mathbf{x}$ at the region level through its centroid, and the second explains the remaining local variation. Moreover, we qualitatively find that the decomposition is often causally interpretable as well. Centroid interventions (Eq. 14) often promote a broad semantic theme, while local offsets (Eq. 15) can produce more fine-grained shifts within the broader theme of the region. We provide examples in Table 1 for the broad and narrow Gaussians shown in Figure 2 and more in §F.2.

**Interpretability of Decomposition**  We evaluate whether the decompositions by MFA and SAEs yield features that are coherent and human-interpretable, rather than artifacts of their training objectives. To this end, we take for each method 50 activations from The Pile (Gao et al., 2020) and decompose them into parts as previously described. Then, we label each feature as interpretable or not in the context of the decomposition based on a reference feature description.

MFA centroids are described as in §4. For the local term $W\hat{\mathbf{z}}$, we do not interpret individual loadings in isolation, as a single direction does not necessarily correspond to a single concept. Instead, meaning is captured by the *subspace as a whole* and the local coordinate system it defines. Therefore, we label relative features as interpretable by asking NLP graduate student annotators to judge whether $\hat{\mathbf{z}}_k$ places $\mathbf{x}$ in a coherent local cluster in the Gaussian's latent space. We compare $\mathbf{x}$'s nearest neighbors to a within-component contrast set of farthest points, and mark it interpretable if the shared concept is strong among neighbors but absent (or much weaker) in the contrast set.

For SAEs, we use feature descriptions from Neuronpedia (Lin, 2023). We define an SAE feature as interpretable if its description relates to the activation context. Since SAEs activate numerous features, we use an LLM judge for labeling and validate it with 100 annotations done by NLP graduate students, finding substantial agreement ($\kappa = 0.61$). For prompt see §G.

Let $\hat{\mathbf{x}} = \sum_i \mathbf{v}_i$ be the decomposition of $\mathbf{x}$ by a method, written as a sum of features. For MFA, we have $\mathbf{v}_1 = \boldsymbol{\mu}_k$ and $\mathbf{v}_2 = W_k \hat{\mathbf{z}}_k$, and for SAEs $\mathbf{v}_j = a_j \mathbf{f}_j$ for each active feature $j$. We quantify the *interpretability fraction* (IF) of $\hat{\mathbf{x}}$ using the magnitude of each feature's contribution:

$$\text{IF}(\mathbf{x}) = \frac{\sum_{i \in \mathcal{I}} \|\mathbf{v}_i\|_2}{\sum_i \|\mathbf{v}_i\|_2}, \tag{16}$$

where $\mathcal{I}$ is the set of features labeled interpretable.

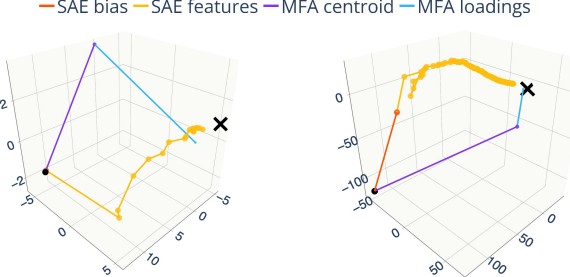

*Figure 4.* **MFA vs. SAE reconstructions.** MFA reconstructs an activation by anchoring it to a region (centroid) and refining it with a region-specific direction, whereas SAEs reconstruct by accumulating many global dictionary features. Left: Llama-3.1-8B, layer 22; right: Gemma-2-2B, layer 18.

*Table 2.* Localization performance of MFA versus unsupervised and supervised baseline methods on RAVEL and MCQA.

| Model | Task | PCA | SAE | MFA | DBM | DAS |
|---|---|---|---|---|---|---|
| Gemma-2 | Continent | 70.0 | 71.7 | 85.7 | 69.7 | 84.2 |
| | Language | 56.0 | 58.9 | 64.0 | 58.0 | 69.7 |
| | Country | 53.0 | 56.1 | 60.0 | 65.0 | 67.7 |
| | MCQA | 77.9 | 64.9 | 80.2 | 82.1 | 92.0 |
| Llama-3.1 | Continent | 74.0 | 70.6 | 81.6 | 78.1 | 81.1 |
| | Language | 57.0 | 56.8 | 67.3 | 63.1 | 69.1 |
| | Country | 54.0 | 57.6 | 62.8 | 60.4 | 64.1 |
| | MCQA | 74.3 | 65.6 | 70.5 | 75.0 | 91.7 |

Across all settings, MFA achieves an average IF of $0.96 \pm 0.2$, indicating that most of the high-contribution features in its decomposition are interpretable, compared to $0.29 \pm 0.2$ for SAEs. This indicates that MFA decomposes activations into a small set of interpretable features, whereas SAEs rely on many features, most of which are not directly interpretable from their context.

## 6. Evaluations

We evaluate MFA on localization and steering benchmarks, where it surpasses both unsupervised and supervised methods on localization and often outperforms SAEs on steering.

### 6.1. Localization

**Experiment** We evaluate MFA on the MIB benchmark (Mueller et al., 2025), using the published code for the MCQA (Wiegreffe et al., 2025) and RAVEL (Huang et al., 2024) settings. Both settings test a method's ability to isolate a *causal variable* in the model's computation and manipulate its behavior by intervening on that variable's representation. MCQA focuses on a *positional pointer* variable and tests if interventions reliably change the model's answers on

multiple-choice questions, whereas RAVEL targets *entity-level* causal variables (Continent, Country, and Language).

We train MFA using the MIB training split and report results on the validation split. For causal localization, we utilize Desiderata-Based Masking (DBM) (De Cao et al., 2020; 2022; Csordás et al., 2021; Davies et al., 2023) on top of MFA's components, mirroring the way SAEs are evaluated on the benchmark. DBM learns a sparse mask over a method's learned basis, selecting a sparse set that best aligns with the benchmark's causal variable. The method is then evaluated using only the chosen basis vectors.

We compare MFA to existing methods: PCA, SAEs (Bricken et al., 2023; Cunningham et al., 2023), DBM and DAS (Geiger et al., 2023), whose scores were taken directly from Mueller et al. (2025). We also ablate MFA on Gemma-2-2B to identify whether the causal variables reside in the loadings or centroids. To this end, we restricted DBM's candidate set to the centroids which removes its ability to utilize the loadings. For additional details see §E.

**Results** Table 2 report accuracy on RAVEL and MCQA. On RAVEL, MFA outperforms PCA and SAEs by large margins (3-16 points) and beats DBM in 5 out of 6 cases. Moreover, MFA performs better on the Continent task than DAS, the current state-of-the-art supervised method and for Llama-3.1-8B comes within two points on the rest of the tasks. For MCQA, MFA outperforms SAEs by up to 15 points and on Gemma, it also exceeds PCA and nearly matches DBM. Complete layerwise results exhibit similar trends and are provided in §E. Inspecting the ablation results, we find that utilizing only the centroids for RAVEL maintains performance (Continent 86, Language 64, Country 59), indicating that these variables are captured primarily by regional information. In contrast, the same restriction substantially degrades MFA performance on MCQA (80%→39%), suggesting that more fine-grained variables require within-region variation that cannot be recovered from global position alone.

Overall, MFA performs strongly on causal localization, consistently improving over the unsupervised baselines, while often being competitive with supervised methods. Moreover, both the centroids and local covariance structures are important for isolating causal variables, with some only showing up as local variation.

### 6.2. Causal Steering

**Experiment** We benchmark MFA against SAEs and supervised difference-in-means (DiffMeans; Rimsky et al., 2024; Marks & Tegmark, 2024; Turner et al., 2024; Singh et al., 2024), ReFT-R1 (Wu et al., 2025), and prompting. For MFA, we intervene with the centroids (Eq. 14). For SAEs, DiffMeans, and ReFT-R1, we use the standard addi-

tive intervention, adding $\alpha$ times the corresponding feature direction to the residual stream, as this was found most effective by Wu et al. (2025) and ablated in §E. For ReFT-r1, we use the provided 500 concepts and respective training data from Wu et al. (2025), and train the ReFT-r1 directions in the same manner as outlined in the paper for the models and layers used in our evaluations. For prompting, since we utilize a completion model, we prepend the instruction "In the following sentences I will discuss the concept: [c]." to the base prompt "I think that", and evaluate the generated continuation following this prefix.

For evaluation, we follow Wu et al. (2025), using the same prompts and 0–2 scoring rubric. Interventions are applied during model inference on the prompt: "<BOS> I think that". We sweep 15 values of the intervention coefficient $\alpha$, and sample 8 completions per value. We then use GPT-4o-mini (OpenAI et al., 2024) to rate each completion on two axes: a *concept score* (alignment with the target concept) and *fluency* (coherence preservation). As in Wu et al. (2025), we report for each centroid/feature the highest scoring set of completions across the sweep and aggregate the concept and fluency scores with a harmonic mean as the final score. As some SAE features suppress rather than promote a concept, we also intervene with the negation, and report the better performing sign for each feature when aggregating over $\alpha$. We provide steering parameters, intervention method ablations and the concept/fluency scores – where we see MFA has signficantly higher conceptual alignment – in §E.

To calculate the concept score, we provide a concept description for each centroid/feature. For MFA, we use the same procedure as in §4. For SAEs, we use descriptions from Neuronpedia (Lin, 2023), which are generated based on the max-activating samples for each feature (Bills et al., 2023; Bricken et al., 2023; Choi et al., 2024; Paulo et al., 2024). DiffMeans, ReFT-r1 and prompting are supervised and have labels by construction.

We apply this evaluation to Llama-3.1-8B and Gemma-2-2B across two layers each for 250 randomly sampled features/-centroids. We compare the trained MFAs of three scales from §4 and the state-of-the-art publicly available SAEs, Llamascope and Gemmascope (He et al., 2024; Lieberum et al., 2024). To train DiffMeans, we use the provided SAE descriptions as target concepts and generate 72 activating and 72 neutral examples per feature. We then calculate the feature vector using the difference of the average token representation for each set (Wu et al., 2025).

**Results**  Figure 5 presents the results, showing that MFA outperforms SAEs and DiffMeans across most settings. On Gemma-2-2B it roughly doubles the median score, and on Llama-3.1-8B it improves the median by about one third,

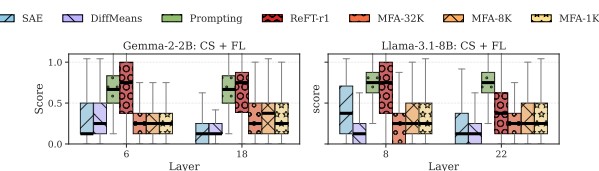

*Figure 5.* Steering results across layers in Gemma-2-2B and Llama-3.1-8B of state-of-the-art SAEs, DiffMeans and 1K, 8K, 32K Gaussian MFAs. Across the majority of settings MFA significantly outperforms DiffMeans and SAEs.

with Layer 8 bringing the average down as an exception. Notably, we observe no consistent gains from increasing MFA's capacity. This suggests that higher capacity shifts centroids toward more specific concepts that remain well described by the Gaussian's high-likelihood contexts. This aligns with §4, where complex structures are represented by multiple Gaussians, hinting that increasing capacity may primarily split broad concepts into additional Gaussians. Consistent with prior work (Wu et al., 2025), ReFT-r1 and prompting outperform all unsupervised methods. Nevertheless, MFA improves substantially over existing unsupervised baselines, narrowing the gap between unsupervised and supervised approaches. These results highlight absolute positions learned by the centroids as an effective unit for steering, driven by higher concept scores, it significantly improves over both SAEs and DiffMeans in the majority of settings.

## 7. Related Work

**Geometric Structures in LMs**  Early work showed that language representations have rich geometric structure, including linear relations (Mikolov et al., 2013a;b; Levy & Goldberg, 2014; Pennington et al., 2014; Arora et al., 2016; Smilkov et al., 2016, *inter alia*). Later analyses showed that contextual LM geometry is more structured and context-dependent, with token embeddings forming usage-specific regions rather than a single global linear space (Coenen et al., 2019a; Ethayarajh, 2019). More recent work has found that modern LM representations span many directions globally but have low intrinsic dimension locally, consistent with a manifold hypothesis Lee et al. (2025); Saglam et al. (2025). Motivated by these results, we use MFA to model LM activation space as a collection of local regions and their low-dimensional directions of variation, enabling scalable feature discovery grounded in local geometry. To our knowledge, our work is the first to apply MFA to LMs.

**Feature Discovery in LMs**  SAEs have become the predominant approach for unsupervised activation decomposition in LMs (Bricken et al., 2023; Cunningham et al., 2023), assuming activations are well modeled by a global sparse dictionary of directions. This view has limitations because it treats directions as the basic unit, despite evidence

that meaningful variation is often multi-dimensional and local (Lee et al., 2025; Saglam et al., 2025; Engels et al., 2025). Unlike SAEs, we tackle decomposition by explicitly modeling activation space as locally organized. We learn a collection of regions with low-dimensional structure, using subspaces as the basic unit of analysis.

This view is further supported by recent work moving beyond single directions to subspace-level structure. Sun et al. (2025) learned concept-conditioned low-rank subspaces for causal intervention, Huang & Hahn (2025) proposed an unsupervised objective that partitions representation space into multiple subspaces, and Tiblias et al. (2025) identified task-relevant feature manifolds with measurable causal effects. These results motivate subspaces as a natural unit of interpretation, but they are either learned via concept- or task-specific objectives or lack a shared, scalable representation of local geometry. We instead learn a single model of locally organized representation space that provides a common basis for decomposition, localization, and steering.

## 8. Conclusion

We propose MFA as a generative model of an LM's activation space, decomposing it into a mixture of low-rank Gaussian regions and their local axes of variation. This local geometric view yields interpretable components that can represent complex, nonlinear structures beyond what a single global set of directions can express. MFA offers a scalable approach to model control that generalizes across layers and models. On recent benchmarks, it not only surpasses existing unsupervised methods but also remains competitive with supervised ones, often exceeding their performance. From a high-level view, our work introduces a new approach for practical activation decomposition in LMs, relying on local geometry rather than a dictionary of isolated directions. We release our code and 12 trained MFAs for Gemma-2-2B and Llama-3.1-8B to facilitate further community research.

## Impact Statement

This work introduces a local-geometry framework for activation decomposition in LMs. Instead of modeling activations as combinations of isolated global directions, we identify regions and local low-rank structure that explains how activations vary locally. This yields better localization of where a feature "lives" in representation space and more precise interventions that target either the region or specific within-region variations. These tools are potentially dual-use: the same ability to isolate and manipulate internal mechanisms could potentially be used to evade safety measures or amplify undesirable behaviors. We highlight this risk while presenting the method for its intended purpose—transparency and controllable, interpretable behavior.

## Acknowledgments

This work was supported in part by a grant from Coefficient Giving, the Academic Research Program at Google, Len Blavatnik and the Blavatnik Family foundation, and the Israel Science Foundation grant 1083/24.

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

*Table 3.* **Convergence and centroid diversity across initializations.** We report the convergence rate in number of gradient descent steps and the mean $\pm$ std of pairwise Euclidean distances between learned component means ($\boldsymbol{\mu}$).

| Initialization | Steps | Mean **pairwise dist.** ($\pm\ std$) |
|---|---|---|
| Random Point | 26136 | $143.76 \pm 106.18$ |
| K-Means | 28008 | $108.26 \pm 65.41$ |
| Random Init | 38880 | $71.92 \pm 10.33$ |

## A. MFA Initialization and Training

While we primarily initialize with K-Means, we also tried two simpler alternatives: (1) fully random initialization and (2) initializing each component from random data points.

**Random Initialization**    We initialize each component mean by sampling from an isotropic normal distribution: $\boldsymbol{\mu}_k \sim \mathcal{N}(\mathbf{0}, \sigma^2 I)$. All other parameters are initialized the same as described in §3.

**Random Point Initialization**    We initialize each component mean by sampling activations uniformly from the dataset. Let $\{\mathbf{x}_i\}_{i=1}^N$ be the corpus of activations and $K$ the number of components. We sample indices $i_1, \ldots, i_K$ uniformly at random and set

$$\boldsymbol{\mu}_k \leftarrow \mathbf{x}_{i_k} \qquad \text{for } k = 1, \ldots, K. \tag{17}$$

All other parameters are initialized the same as described in §3. Additionally, we note that future work may make this initialization more robust by resampling Gaussians who are initialized close in activation space, thus motivating diverse Gaussians.

**Initialization Comparison**    To compare initialization strategies, we train MFAs with identical hyperparameters while varying only the initialization method (K-Means, random, and random point). We measure convergence efficiency as the number of training iterations until convergence, defined as when the change in log-likelihood between successive iterations falls below $10^{-3}$.

To assess whether the learned components remain diverse after training, we compute pairwise Euclidean distances between component means $\{\boldsymbol{\mu}_k\}_{k=1}^K$.

**results**    We present the results in 3. Fully random initialization often converges to poor solutions, with little variation in the pairwise euclidean distance and relatively higher NLL. Random point initialization scales to very large datasets and converges fast, but is more prone to local minima as is noticeable by the higher NLL compared to K-Means. Additionally, we see the variance is much higher than K-Means, indicating a less uniform spread of centroids. Lastly, K-Means converges fast, but its initialization is slower and less scalable than random point initialization. However, it yields better coverage of the dataset and converges to a better minima.

## B. Annotation Statistical Testing

In this section we provide details on the statistical testing conducted to validate the LLMs capacity to label both the loadings and the Gaussians in §4. We provide prompts and annotation instructions in §G.

**Method**    First, to identify consistency between human and LLM annotators we examine agreement using Cohen's $\kappa$ (i) between humans and (ii) between the LLM and each human.

Second, we use the Alternative Annotator Test (Calderon et al., 2025). We compare how well the LLM agrees with the other human annotators versus how well each individual human agrees with the others (leave-one-out). Following the procedure outlined in Calderon et al. (2025) we compute the winning rate ($\omega$, the fraction of humans for whom the LLM is judged better) and deem the LLM replaceable if $\omega \geq 0.5$ (with $q = 0.05$). For all tasks we use $\epsilon = 0.1$.

For the loading annotation task (which requires interpreting context), labels were provided by a mix of NLP graduate and doctoral students. We ran statistical testing on 58 randomly sampled loadings, using the annotation instructions in §G and

the procedure described in §4. For the gaussian annotation task we had non-expert human annotators annotate 29 samples with the procedure outlined in §4.

**Results**    For the loading task we found that across 58 sampled loadings, human annotators showed moderate agreement (mean Cohen's $\kappa = 0.29$ across pairs). The LLM matched each human annotator at least as well, with higher agreement on average (mean $\kappa = 0.44$). Using the alternative annotator test (leave-one-out with BY correction, $q = 0.05$), the LLM significantly outperformed the excluded human in all three comparisons, yielding a winning rate of $\omega = 1.0$ and therefore meeting the replacement criterion ($\omega \geq 0.5$).

For the Gaussian labeling task across the sampled Gaussians, the three human annotators agreed perfectly (pairwise Cohen's $\kappa = 1.0$). The LLM annotations also matched each human perfectly (mean $\kappa = 1.0$). Under the alternative annotator test (leave-one-out with BY correction, $q = 0.05$), the LLM met the replacement criterion ($\omega = 1.0 \geq 0.5$), indicating it is statistically indistinguishable from the human annotators on this task.

## C. Labeling Loadings

For a single activation $\mathbf{x}$ assigned to component $k$, we therefore test whether its latent coordinates $\hat{\mathbf{z}}_k$ place it in the *right concept group* within that region. We retrieve the 10 nearest neighbors to $\hat{\mathbf{z}}_k$ in the low-dimensional latent space defined by the Gaussian, along with a contrast set given by the 10 farthest points in the same space. This within-component contrast isolates what the subspace is separating inside the region. We then judge whether $\mathbf{x}$ shares a coherent concept with its nearest neighbors. We also require this concept to be absent or much weaker in the contrast set. If so, we treat the placement induced by $\hat{\mathbf{z}}_k$ as an interpretable account of the local refinement.

## D. Reconstruction Analysis

**Experiment**    To evaluate whether a local low-rank factor model is a reasonable approximation of activations, we measure reconstruction error on a held-out validation set. We compare MFA with SAEs as a baseline, using 10 million activations sampled from Wikipedia. For each activation $\mathbf{x} \in \mathbb{R}^d$, MFA reconstructs $\hat{\mathbf{x}}$ using Eq. 11, while the SAE reconstructs via encoding then decoding. We report the mean squared reconstruction error (MSE) over all samples:

$$\text{MSE} \;=\; \frac{1}{Nd} \sum_{i=1}^{N} \|\mathbf{x}_i - \hat{\mathbf{x}}_i\|_2^2, \tag{18}$$

along with the standard error of this estimate to indicate measurement precision. We measure MSE for the 12 MFAs of Gemma-2-2B and Llama-3.1-8B from §4. For SAEs we evaluate Gemmascope-65k (Lieberum et al., 2024) and Llamascope (He et al., 2024) at the same layers of the MFAs. In addition, we evaluate a weak K-means baseline that reconstructs a given activation by assigning it to its closest centroid.

**Results**    Table 4 shows that reconstruction quality improves with the number of MFA components $K$. Increasing $K$ from 1K to 8K yields a substantial reduction in error, while the additional improvement from 8K to 32K is comparatively lower, indicating diminishing returns once $K$ is sufficiently large for the corpus. Across all settings, SAEs achieve lower reconstruction error. This is expected because SAEs allow a more flexible, sample-specific reconstruction. Each activation can be expressed as a sparse combination of dictionary elements whose support can change across inputs. MFA instead commits to a single component and reconstructs within a fixed rank $R$ subspace around that centroid, which makes the reconstruction less expressive with the used $R = 10$. The remaining gap is therefore consistent with the rank constraint. Importantly, MFA substantially outperforms the K-means baseline, which consistently has $1.3 - 1.5$ times higher MSE. This indicates that modeling within-region variation with a learned low-rank structure captures a considerable portion of the structured representation of the activation.

**Limitation**    A key limitation of MFA is that it explicitly models the activation distribution it is trained on. This means that the centroids and low-rank subspaces model the regions of the activation space that occur in the training corpus. As a result, when an activation lies in a region that is rare or out of distribution of the training set (not part of the Gaussians we model), MFA may assign it to the nearest available component even if none provides a good local fit. In these cases, MFA yields high reconstruction error. This failure mode is inherent to the model class. Increasing $K$ can expand coverage of the training

*Table 4.* Reconstruction error reported as MSE for MFAs, SAEs, and a K-Means baseline. For the baseline we additionally provide the error ratio with MFA of the same size. All results were found to be statistically significant with standard error $< 1e - 4$.

| Model | Layer | SAE | MFA-1K | KMeans-1K | MFA-8K | KMeans-8K | MFA-32K | KMeans-32K |
|---|---|---|---|---|---|---|---|---|
| Gemma-2 | 6 | 0.592 | 1.098 | 1.530 ($\times$1.4) | 0.896 | 1.265 ($\times$1.4) | 0.800 | 1.104 ($\times$1.4) |
| | 18 | 4.595 | 5.430 | 7.629 ($\times$1.4) | 4.678 | 6.622 ($\times$1.4) | 4.477 | 5.966 ($\times$1.3) |
| Llama-3.1 | 8 | 0.023 | 0.005 | 0.007 ($\times$1.3) | 0.004 | 0.006 ($\times$1.4) | 0.004 | 0.005 ($\times$1.4) |
| | 22 | 0.038 | 0.064 | 0.086 ($\times$1.3) | 0.051 | 0.074 ($\times$1.5) | 0.045 | 0.066 ($\times$1.5) |

distribution, but it does not guarantee low error on regions that were never observed. Given this, we still find MFA isolates meaningful and useful features that generalize to tasks like steering and localization.

## E. Benchmarking: Additional Details

**Localization (MCQA and RAVEL).** We train MFAs for the localization experiments on both gemma-2-2b and Llama-3.1-8b for all layers. We use the train sets of the Country, Language, and Continent tasks combined. We fit an MFA independently per layer with $K \in \{100, 250\}$ for MCQA and $K \in \{25, 50\}$ for RAVEL. We sweep ranks $R \in \{25, 50\}$ for MCQA and $R \in \{10, 20\}$ for RAVEL. MFAs are trained for 400 epochs with batch size 256 and learning rate $10^{-3}$. For dBM, we use a routed variant better suited to mixture models: the MFA encoder first selects the most likely Gaussian component, and dBM learns a binary mask over the resulting routed feature representation, consisting of the selected component identity and its local latent coordinates. In the paper, we report the results for the best combination over the parameters. For MIB parameters, we use the default parameters provided in their codebase. We report the full layerwise results for RAVEL in Tables 5 to 10, and for MCQA in Tables 11 and 12.

Additionally, to isolate whether using a mixture model provides additional benefits beyond the use of noise modeling in MFA, we added a global Factor Analysis (FA) baseline. Notably, this baseline was only applied for localization as it does not extend to our full large-scale steering setting, where a single global FA is fundamentally limited. The number of features it can represent is bounded by the activation dimension, which is too restrictive for the scale we used of 100 million activations. In that regime, the mixture is not merely helpful but necessary. We trained FA with the same rank as the corresponding MFA to isolate the added value of the mixture structure. We found that on RAVEL, a single FA matched MFA having the same score. However, on MCQA it reached only 35.0% accuracy for Gemma and 34.8% for Llama, substantially below MFA with 80.2% and 70.5%. This is consistent with our broader finding that RAVEL requires substantially lower rank than MCQA, and further supports the need for locality in more complex settings.

| | 0 | 1 | 2 | 3 | 4 | 5 | 6 | 7 | 8 | 9 | 10 | 11 | 12 | 13 | 14 | 15 | 16 | 17 | 18 | 19 | 20 | 21 | 22 | 23 | 24 | 25 |
|---|---|---|---|---|---|---|---|---|---|---|---|---|---|---|---|---|---|---|---|---|---|---|---|---|---|---|
| MFA | 74.3 | 77.5 | 80.1 | 82.4 | 76.3 | 77.1 | 68.2 | 68.6 | 56.0 | 69.5 | 71.7 | 73.5 | 71.0 | 75.1 | 74.2 | 78.9 | 82.0 | 82.6 | 84.5 | 84.9 | 83.9 | 84.6 | 85.5 | 85.4 | 85.4 | 85.4 |
| SAE | 44.8 | 47.4 | 54.9 | 54.5 | 60.0 | 60.4 | 62.6 | 66.3 | 66.7 | 68.1 | 66.9 | 66.3 | 63.9 | 66.3 | 63.1 | 65.6 | 71.7 | 70.2 | 68.0 | 68.9 | 66.1 | 69.2 | 65.9 | 64.1 | 62.4 | 66.4 |
| DBM | 58.4 | 58.0 | 59.8 | 60.8 | 60.6 | 61.3 | 63.0 | 65.9 | 69.0 | 69.0 | 68.6 | 69.1 | 69.1 | 68.1 | 69.6 | 64.7 | 68.9 | 68.7 | 68.5 | 69.7 | 68.4 | 67.5 | 67.9 | 65.8 | 65.2 | 61.8 |

*Table 5.* RAVEL continent results for Gemma-2 2B

| | 0 | 1 | 2 | 3 | 4 | 5 | 6 | 7 | 8 | 9 | 10 | 11 | 12 | 13 | 14 | 15 | 16 | 17 | 18 | 19 | 20 | 21 | 22 | 23 | 24 | 25 | 26 | 27 | 28 | 29 | 30 | 31 |
|---|---|---|---|---|---|---|---|---|---|---|---|---|---|---|---|---|---|---|---|---|---|---|---|---|---|---|---|---|---|---|---|---|
| MFA | 80.0 | 81.0 | 79.0 | 77.0 | 80.0 | 77.0 | 76.0 | 57.0 | 62.0 | 78.0 | 77.0 | 74.0 | 77.0 | 77.0 | 81.0 | 81.0 | 81.0 | 82.0 | 82.0 | 82.0 | 82.0 | 82.0 | 81.0 | 82.0 | 82.0 | 82.0 | 82.0 | 82.0 | 82.0 | 82.0 | 82.0 | 82.0 |
| SAE | 54.4 | 48.0 | 57.2 | 56.1 | 58.9 | 61.3 | 61.2 | 59.9 | 63.7 | 62.3 | 63.2 | 63.6 | 66.6 | 63.5 | 59.8 | 61.6 | 67.3 | 70.5 | 70.6 | 70.6 | 68.8 | 67.9 | 66.7 | 63.6 | 62.4 | 62.6 | 62.6 | 62.4 | 62.4 | 62.6 | 62.6 | 62.6 |
| DBM | 58.2 | 60.1 | 59.9 | 59.2 | 60.6 | 63.3 | 64.3 | 66.3 | 66.2 | 65.5 | 67.4 | 68.1 | 68.0 | 67.1 | 65.0 | 66.5 | 74.2 | 73.8 | 73.9 | 77.3 | 76.2 | 78.1 | 74.8 | 77.3 | 76.1 | 73.3 | 72.1 | 71.6 | 68.0 | 67.9 | 66.1 | 64.7 |

*Table 6.* RAVEL continent results for Llama-3.1 8B

| | 0 | 1 | 2 | 3 | 4 | 5 | 6 | 7 | 8 | 9 | 10 | 11 | 12 | 13 | 14 | 15 | 16 | 17 | 18 | 19 | 20 | 21 | 22 | 23 | 24 | 25 |
|---|---|---|---|---|---|---|---|---|---|---|---|---|---|---|---|---|---|---|---|---|---|---|---|---|---|---|
| MFA | 48.4 | 53.5 | 54.5 | 58.0 | 52.6 | 52.1 | 42.7 | 47.8 | 41.2 | 43.1 | 43.7 | 46.1 | 46.2 | 50.0 | 46.7 | 56.1 | 57.6 | 56.8 | 57.5 | 58.4 | 58.3 | 58.9 | 59.8 | 59.6 | 59.6 | 59.6 |
| SAE | 34.0 | 36.4 | 42.4 | 39.1 | 46.8 | 46.6 | 50.4 | 51.5 | 52.8 | 53.5 | 50.8 | 49.9 | 50.4 | 48.6 | 48.9 | 55.7 | 56.1 | 53.1 | 49.1 | 48.7 | 49.0 | 49.3 | 50.4 | 50.4 | 50.5 | 50.6 |
| DBM | 45.1 | 47.2 | 48.2 | 49.0 | 48.9 | 49.6 | 52.3 | 52.9 | 56.0 | 56.9 | 55.6 | 55.0 | 54.5 | 54.7 | 54.6 | 63.1 | 65.0 | 65.0 | 51.2 | 53.4 | 51.1 | 50.2 | 50.4 | 50.3 | 50.3 | 50.6 |

*Table 7.* RAVEL country results for Gemma-2 2B

**Causal.** Causal steering is performed on `gemma-2-2b` (layers 6 and 18) and `Llama-3.1-8b` (layers 8 and 22). For MFA, we apply interpolation-based interventions and sweep $\alpha \in$

| | 0 | 1 | 2 | 3 | 4 | 5 | 6 | 7 | 8 | 9 | 10 | 11 | 12 | 13 | 14 | 15 | 16 | 17 | 18 | 19 | 20 | 21 | 22 | 23 | 24 | 25 | 26 | 27 | 28 | 29 | 30 | 31 |
|---|---|---|---|---|---|---|---|---|---|---|---|---|---|---|---|---|---|---|---|---|---|---|---|---|---|---|---|---|---|---|---|---|
| MFA | 60.0 | 61.0 | 59.0 | 56.0 | 57.0 | 54.0 | 51.0 | 45.0 | 47.0 | 54.0 | 53.0 | 51.0 | 53.0 | 56.0 | 60.0 | 60.0 | 62.0 | 63.0 | 63.0 | 63.0 | 63.0 | 63.0 | 62.0 | 63.0 | 63.0 | 63.0 | 63.0 | 63.0 | 63.0 | 63.0 | 63.0 | 63.0 |
| SAE | 45.0 | 38.6 | 45.5 | 40.6 | 47.6 | 47.4 | 49.5 | 50.9 | 52.0 | 49.9 | 49.4 | 50.1 | 51.7 | 52.4 | 57.6 | 56.1 | 53.7 | 51.8 | 52.2 | 55.0 | 55.1 | 55.9 | 54.3 | 54.7 | 50.4 | 50.9 | 50.5 | 50.5 | 50.5 | 50.4 | 50.5 | 50.7 |
| DBM | 49.2 | 48.2 | 45.6 | 49.1 | 48.9 | 48.9 | 50.6 | 52.1 | 52.6 | 54.2 | 55.1 | 55.3 | 56.5 | 58.2 | 59.8 | 60.4 | 56.3 | 58.3 | 58.3 | 57.8 | 56.4 | 57.2 | 55.4 | 54.9 | 50.5 | 51.5 | 50.4 | 50.4 | 50.5 | 50.5 | 50.6 | 50.7 |

*Table 8.* RAVEL country results for Llama-3.1 8B

| | 0 | 1 | 2 | 3 | 4 | 5 | 6 | 7 | 8 | 9 | 10 | 11 | 12 | 13 | 14 | 15 | 16 | 17 | 18 | 19 | 20 | 21 | 22 | 23 | 24 | 25 |
|---|---|---|---|---|---|---|---|---|---|---|---|---|---|---|---|---|---|---|---|---|---|---|---|---|---|---|
| MFA | 57.2 | 58.7 | 61.1 | 61.9 | 56.2 | 55.4 | 48.6 | 47.8 | 44.5 | 50.2 | 48.6 | 52.8 | 54.2 | 54.9 | 54.1 | 58.2 | 60.5 | 61.1 | 62.4 | 62.9 | 63.4 | 63.1 | 63.2 | 63.1 | 63.0 | 63.0 |
| SAE | 50.8 | 43.1 | 49.9 | 44.0 | 50.9 | 48.6 | 50.8 | 53.7 | 54.3 | 53.1 | 53.7 | 53.0 | 53.3 | 52.8 | 53.0 | 52.9 | 53.4 | 55.7 | 58.9 | 57.0 | 56.0 | 56.3 | 54.8 | 54.6 | 54.6 | 54.6 |
| DBM | 53.4 | 54.1 | 54.0 | 53.8 | 53.0 | 53.5 | 54.1 | 53.6 | 53.8 | 54.4 | 53.8 | 54.3 | 54.3 | 53.4 | 54.1 | 53.8 | 51.6 | 53.3 | 58.0 | 56.8 | 55.8 | 56.2 | 54.6 | 54.6 | 54.7 | 54.6 |

*Table 9.* RAVEL language results for Gemma-2 2B

| | 0 | 1 | 2 | 3 | 4 | 5 | 6 | 7 | 8 | 9 | 10 | 11 | 12 | 13 | 14 | 15 | 16 | 17 | 18 | 19 | 20 | 21 | 22 | 23 | 24 | 25 | 26 | 27 | 28 | 29 | 30 | 31 |
|---|---|---|---|---|---|---|---|---|---|---|---|---|---|---|---|---|---|---|---|---|---|---|---|---|---|---|---|---|---|---|---|---|
| MFA | 65.0 | 65.0 | 64.0 | 63.0 | 63.0 | 57.0 | 58.0 | 46.0 | 50.0 | 57.0 | 59.0 | 58.0 | 60.0 | 62.0 | 63.0 | 63.0 | 66.0 | 67.0 | 67.0 | 67.0 | 67.0 | 67.0 | 67.0 | 67.0 | 67.0 | 67.0 | 67.0 | 67.0 | 67.0 | 67.0 | 67.0 | 67.0 |
| SAE | 50.2 | 52.2 | 48.3 | 53.0 | 54.2 | 54.3 | 54.2 | 52.6 | 54.5 | 54.8 | 54.9 | 55.4 | 55.5 | 55.2 | 55.3 | 53.9 | 55.1 | 56.5 | 56.0 | 55.4 | 55.6 | 56.8 | 55.4 | 55.7 | 55.6 | 55.6 | 55.6 | 55.7 | 55.7 | 55.7 | 55.7 | 55.9 |
| DBM | 54.2 | 53.8 | 54.5 | 53.5 | 54.2 | 54.4 | 55.2 | 56.0 | 56.4 | 56.0 | 56.2 | 56.8 | 57.8 | 57.5 | 57.8 | 59.5 | 63.0 | 63.1 | 61.2 | 61.8 | 62.8 | 61.9 | 59.6 | 60.1 | 56.7 | 56.8 | 55.7 | 55.7 | 55.8 | 55.8 | 55.8 | 55.9 |

*Table 10.* RAVEL language results for Llama-3.1 8B

| | 0 | 1 | 2 | 3 | 4 | 5 | 6 | 7 | 8 | 9 | 10 | 11 | 12 | 13 | 14 | 15 | 16 | 17 | 18 | 19 | 20 | 21 | 22 | 23 | 24 | 25 |
|---|---|---|---|---|---|---|---|---|---|---|---|---|---|---|---|---|---|---|---|---|---|---|---|---|---|---|
| PCA | 52.2 | 46.2 | 47.9 | 48.2 | 45.1 | 42.7 | 41.9 | 48.7 | 50.6 | 49.2 | 50.2 | 51.4 | 50.6 | 52.3 | 48.5 | 50.3 | 36.6 | 77.9 | 63.3 | 68.9 | 70.7 | 59.3 | 60.4 | 61.7 | 59.7 | 67.1 |
| SAE | 50.0 | 41.9 | 53.9 | 47.8 | 44.0 | 52.9 | 49.6 | 51.4 | 49.0 | 50.2 | 49.1 | 47.8 | 46.6 | 46.4 | 45.4 | 47.4 | 44.4 | 64.8 | 60.1 | 54.7 | 57.4 | 57.3 | 54.7 | 54.7 | 50.8 | 54.0 |
| MFA | 46.1 | 44.3 | 45.9 | 45.2 | 45.3 | 47.0 | 40.0 | 40.2 | 40.9 | 41.5 | 41.5 | 43.7 | 41.9 | 40.6 | 41.2 | 39.1 | 36.1 | 80.3 | 68.7 | 66.9 | 68.7 | 68.7 | 62.4 | 66.2 | 61.3 | 58.6 |
| DBM | 58.7 | 59.8 | 57.1 | 58.5 | 56.7 | 56.7 | 58.9 | 59.9 | 56.4 | 55.8 | 57.0 | 60.3 | 61.2 | 65.6 | 68.8 | 63.9 | 53.5 | 82.1 | 74.9 | 66.9 | 66.6 | 70.0 | 65.7 | 63.6 | 63.2 | 64.9 |

*Table 11.* MCQA results for Gemma-2 2B

| | 0 | 1 | 2 | 3 | 4 | 5 | 6 | 7 | 8 | 9 | 10 | 11 | 12 | 13 | 14 | 15 | 16 | 17 | 18 | 19 | 20 | 21 | 22 | 23 | 24 | 25 | 26 | 27 | 28 | 29 | 30 | 31 |
|---|---|---|---|---|---|---|---|---|---|---|---|---|---|---|---|---|---|---|---|---|---|---|---|---|---|---|---|---|---|---|---|---|
| PCA | 37.5 | 33.6 | 33.3 | 36.4 | 37.8 | 39.2 | 35.6 | 40.6 | 43.8 | 42.0 | 44.5 | 44.9 | 46.3 | 48.4 | 47.7 | 53.5 | 33.3 | 70.0 | 68.9 | 69.6 | 71.4 | 71.8 | 73.6 | 72.2 | 74.3 | 58.2 | 58.5 | 61.1 | 64.3 | 58.6 | 61.4 | 68.3 |
| SAE | 43.1 | 44.5 | 51.7 | 53.5 | 53.5 | 55.6 | 58.1 | 54.5 | 60.3 | 59.9 | 60.7 | 62.8 | 59.9 | 59.9 | 54.2 | 56.0 | 33.7 | 65.6 | 64.9 | 65.6 | 63.8 | 64.5 | 65.6 | 63.5 | 63.8 | 58.8 | 53.9 | 55.2 | 60.3 | 63.1 | 54.2 | 43.5 |
| MFA | 41.0 | 53.3 | 46.3 | 52.5 | 47.1 | 53.1 | 56.0 | 61.4 | 59.1 | 56.0 | 55.3 | 56.8 | 59.8 | 61.5 | 34.0 | 65.4 | 65.2 | 69.7 | 68.2 | 69.0 | 68.2 | 69.0 | 69.8 | 68.9 | 70.6 | 66.7 | 65.2 | 71.4 | 67.6 | 58.2 | | |
| DBM | 57.4 | 61.0 | 59.9 | 57.8 | 60.3 | 63.5 | 64.6 | 67.5 | 70.7 | 69.7 | 68.2 | 71.1 | 70.0 | 67.1 | 63.9 | 61.7 | 41.9 | 72.5 | 70.7 | 75.0 | 74.0 | 73.2 | 72.2 | 73.6 | 69.6 | 67.1 | 65.6 | 66.0 | 65.3 | 64.9 | 67.1 | 66.5 |

*Table 12.* MCQA results for Llama-3.1 8B

$\{0.15, 0.20, 0.25, 0.30, 0.325, 0.35, 0.375, 0.425, 0.45, 0.475, 0.50, 0.55, 0.60, 0.80\}$.

For SAE baselines, we use Gemmascope 65k (Lieberum et al., 2024) and Llamascope 131k (He et al., 2024), randomly sampling 250 features per layer. We report additive interventions (more effective in our setting) and sweep $\alpha \in \{0.4, 0.8, 1.2, 1.6, 2.0, 3.0, 4.0, 6.0, 8.0, 10.0, 20.0, 40.0, 60.0, 100.0\}$, using the same set of values as in Wu et al. (2025).

For each condition, we generate 8 continuations from the prompt ``I think that'' with max 50 new tokens, top-$k$=30, and top-$p$=0.3. For scoring prompts see Wu et al. (2025) Concept Score prompt and Fluency Prompt.

Additionally, we run ablations on Gemma-2-2B (layer 18) to select the most appropriate intervention scheme (interpolation vs. additive) for both SAEs and MFA on a randomly sampled set of 100 features/Gaussians that was not used for the final results in the paper. For the 32K-MFA, we applied an additive intervention to the centroids using the same $\alpha$ values as above. Performance dropped substantially, with a mean final score of $0.124 \pm 0.14$ opposed to $0.24 \pm 0.20$ with interpolation. This supports the view that MFA centroids primarily encode absolute position in activation space, so additive perturbations are poorly matched to their inductive bias.

We performed the complementary ablation for SAEs, using an interpolation-based intervention with $\alpha \in 0.3, 0.35, 0.4, 0.45, 0.475, 0.5, 0.525, 0.55, 0.6, 0.65, 0.7$. SAE performance degraded sharply, yielding a mean final score of $0.177 \pm 0.19$ as opposed to $0.195 \pm .21$. Furthermore, SAEs are commonly used with additive interventions further supporting the empirical results.

We utilized the best performing intervention method from the ablations for each decomposition.

**Causal Fluency and Concept Scores**  We report here the Concept and Fluency scores for the steering evaluation. Notably, MFA scores significantly higher than SAEs and DiffMeans in concept score and for Gemma-2-2b even matching ReFT-r1 and prompting, indicating it promotes very interpretable concepts that are well described by the high-likelihood samples. Meanwhile, fluency scores are fairly consistent with MFA's fluency on average being slightly lower.

| Model | Method | Early layer | Late layer |
|---|---|---|---|
| Gemma-2-2B | SAE | $0.967 \pm 0.111$ | $1.004 \pm 0.031$ |
| | DiffMeans | $1.002 \pm 0.033$ | $1.003 \pm 0.035$ |
| | Prompting | $1.010 \pm 0.045$ | $1.010 \pm 0.045$ |
| | ReFT-r1 | $0.927 \pm 0.152$ | $0.958 \pm 0.089$ |
| | MFA-32K | $0.841 \pm 0.234$ | $0.855 \pm 0.197$ |
| | MFA-8K | $0.844 \pm 0.224$ | $0.847 \pm 0.196$ |
| | MFA-1K | $0.869 \pm 0.226$ | $0.897 \pm 0.182$ |
| Llama-3.1-8B | SAE | $0.938 \pm 0.138$ | $0.962 \pm 0.130$ |
| | DiffMeans | $0.980 \pm 0.086$ | $0.990 \pm 0.086$ |
| | Prompting | $0.995 \pm 0.028$ | $0.995 \pm 0.028$ |
| | ReFT-r1 | $0.933 \pm 0.153$ | $0.863 \pm 0.203$ |
| | MFA-32K | $0.912 \pm 0.178$ | $0.892 \pm 0.204$ |
| | MFA-8K | $0.902 \pm 0.178$ | $0.904 \pm 0.170$ |
| | MFA-1K | $0.935 \pm 0.149$ | $0.954 \pm 0.127$ |

*Table 13.* Fluency scores computed over all evaluated units, measured at the intervention setting selected by the best final score. We report mean $\pm$ standard deviation for each method, model, and evaluated layer.

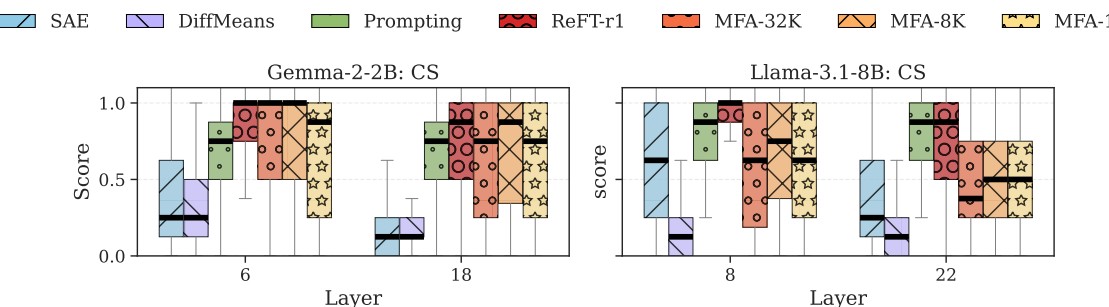

*Figure 6.* Concept Score steering results across layers in Gemma-2-2B and Llama-3.1-8B of state-of-the-art SAEs, DiffMeans and 1K, 8K, 32K Gaussian MFAs. Across all settings MFA significantly outperforms DiffMeans and SAEs. Strongly promoting its associated concepts.

## F. Examples

### F.1. Neighborhood Examples

We provide a range of example Gaussian neighborhoods. For each Gaussian, we annotate a small set of randomly sampled tokens to give a qualitative sense of its content; these tokens are illustrative and may not fully represent the broader concept captured by the component. We show 10 examples in total: 5 from Llama-3.1-8B (layer 22) and 5 from Gemma-2-2B (layer 18).

## Llama-3.1-8B

## Gemma-2-2B

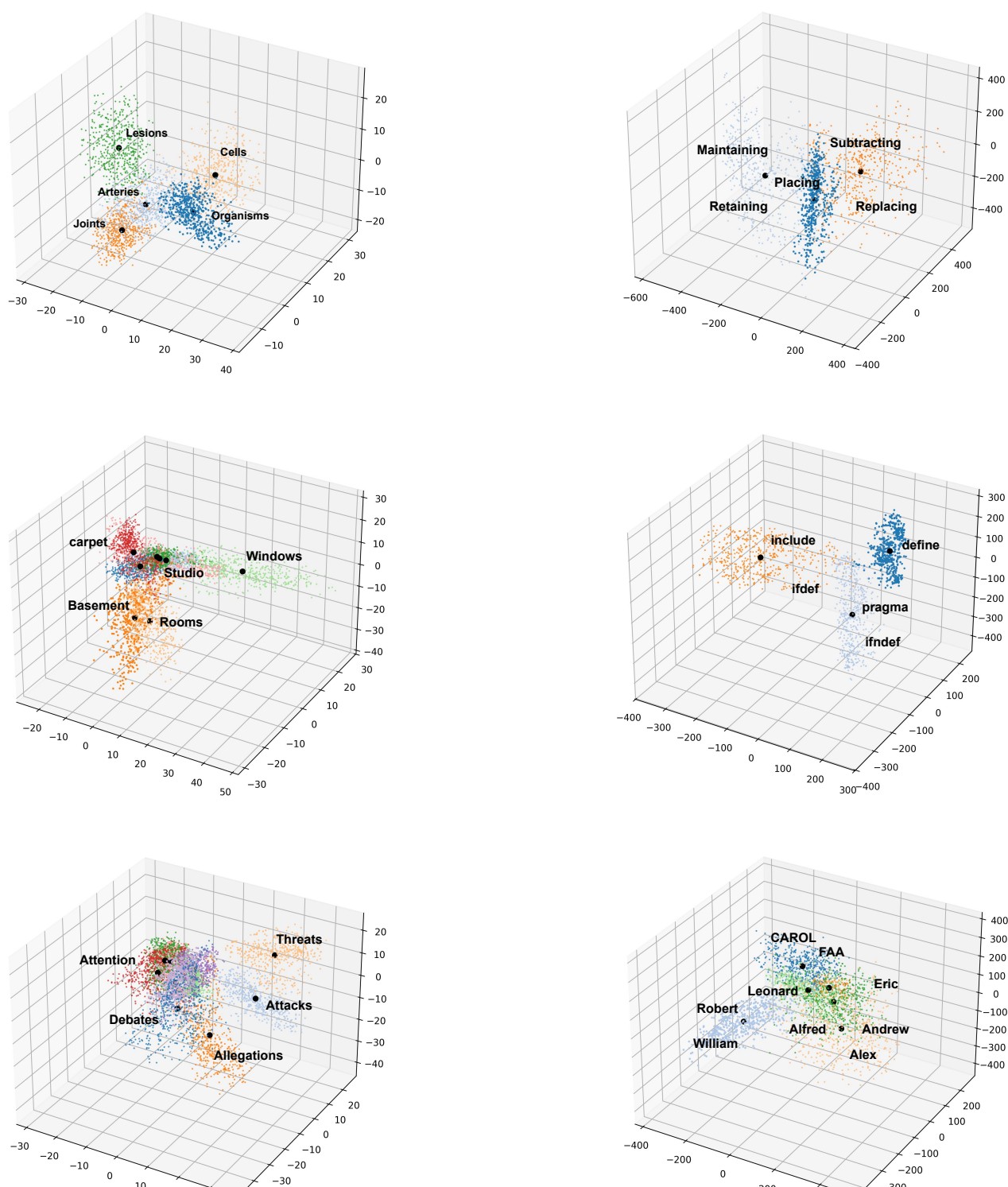

**Llama-3.1-8B (cont.)**                    **Gemma-2-2B (cont.)**

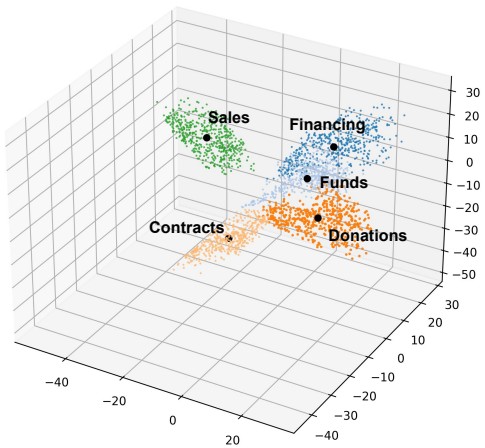  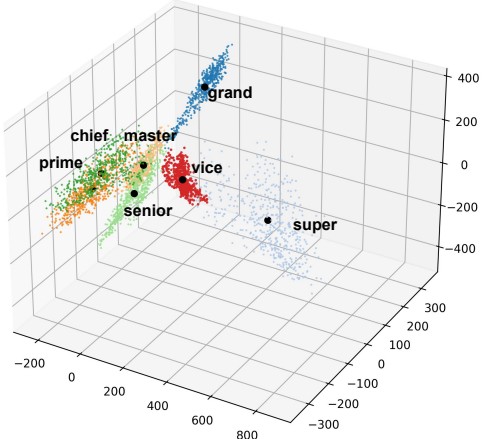

## F.2. Steering Examples

We provide additional examples of steering towards the centroids and independently steering with the local covariance structure. We consistently see that steering with centroids promotes a broad topic that aligns with the cluster it represents. Additionally, intervening with the directions of variation, even from not within the region, often promotes a sub-concept of the broader theme. We see this as a phenomenon where the centroid move changes the activation's absolute position. It pushes the representation into a region that similar contexts tend to pass through, so it reliably brings out the broad topic associated with that Gaussian. In contrast, the local covariance captures how activations vary within the region. Because points in the same region are already very similar, this variation is often especially clean, separating a single semantic attribute and producing a targeted sub-concept shift within the broader theme. This is often the case for narrow Gaussians too, showing that even when the Gaussian capture only very specific tokens or structures, the learned covariance models a wide range of concepts.

We present this as qualitative evidence, since the MFA loadings define the local subspace of variation rather than a unique semantic direction. As a result, stepping along a single loading may not always line up with the dominant direction of variation in that region.

The first table shows examples of top promoted tokens using the centroid intervention (Eq. 14) and loading interventions (Eq. 15) from Llama-3.1-8B layer 22 and the second from Gemma-2-2B layer 18.

*Table 14.* **More centroid vs. loading examples (Llama-3.1-8B).** Each centroid shows broad theme promotion, while loadings isolate sharper subthemes.

| Term | Descriptor | Top promoted tokens |
|---|---|---|
| **(1) Superheroes / comics universe** | | |
| $\mu$ | *superheroes* | Superman, Batman, Joker, Gotham, Deadpool, superhero |
| $w_1$ | *Superman* | Superman, Clark, krypton, rypton, SUPER, restoring |
| $w_2$ | *DC film studio* | Batman, Warner, WB, Nolan, Snyder, films |
| $w_3$ | *Batman* | Joker, Gotham, Commissioner, Mafia, inmates, profiler |
| **(2) Vascular procedures / clinical devices** | | |
| $\mu$ | *catheters, needles* | biopsy, cath, artery, needle, infusion, injection |
| $w_1$ | *angioplasty tools* | balloon, angi, coronary, flexible, tapered, strut |
| $w_2$ | *injection verbs* | injections, injection, injecting, inject, injected, Injection |
| $w_3$ | *connectors, tubing* | adapter, connector, nozzle, hose, cartridge, apparatus |
| **(3) Places / US cities and neighborhoods** | | |
| $\mu$ | *US city names* | Newark, Lexington, Bronx, Omaha, Honolulu, Albany |
| $w_1$ | *East Coast metro* | Hartford, Stamford, Greenwich, Brooklyn, Manhattan, Bronx |
| $w_2$ | *downtown / inner-city* | downtown, Downtown, Harlem, Detroit, inner, ghetto |
| $w_3$ | *suburbs framing* | suburban, suburbs, suburb, Anaheim, Los |
| **(4) Organs / biomedical anatomy** | | |
| $\mu$ | *organs anatomy* | intestine, liver, uterus, sple, pancre, kidneys |
| $w_1$ | *neuroscience terms* | brain, brains, neurons, neuro, neuroscience, cerebral |
| $w_2$ | *genetics / immunity* | antigen, antibody, gene, genome, mouse, vaccine |
| $w_3$ | *organ physiology* | lungs, kidneys, arteries, renal, blood, pulmonary |

*Table 15.* **More centroid vs. loading examples (Gemma-2-2B).** Each centroid shows broad theme promotion, while loadings isolate sharper subthemes. Additionally, we label narrow Gaussians to show that local variation is often meaningful.

| Term | Descriptor | Top promoted tokens |
|------|-----------|---------------------|
| **(1) continents/countries** | | |
| $\mu$ | *continents* | `continent, Europe, EU, France, Parlamento, Countries` |
| $w_1$ | *continent names* | `Europe, Asia, Africa, continents, European` |
| $w_2$ | *Africa* | `Africa, Ebola, malaria, Angola, regions` |
| $w_3$ | *SE Asia* | `Nang, Cambodian, Ceylon, Nepali, Indon` |
| **(2) Sleep / tiredness** | | |
| $\mu$ | *sleep* | `apnea, deprivation, sleep, sleepless` |
| $w_1$ | *sleep disorders* | `apnea, insomnia, deprivation, circadian, disorders` |
| $w_2$ | *sleep lexemes* | `slept, Sleep, sleeps, SLEEP` |
| $w_3$ | *waking / until* | `until, Until, hrs, woken, UNTIL` |
| **(3) Security / secure (Narrow Gaussian)** | | |
| $\mu$ | *secure, affixes* | `safeguard, unlock, Secured, Against, ness, able` |
| $w_1$ | *email security* | `Gmail, gmail, password, inbox, email, browser` |
| $w_2$ | *cryptography* | `encrypted, ciphertext, passphrase, authenticated, smtp` |
| $w_3$ | *fasten / attach* | `attaches, affixed, fastened, fastening, attach` |
| **(4) Rooms / venues** | | |
| $\mu$ | *rooms, interiors* | `room, rooms, foyer, showroom, floor, located` |
| $w_1$ | *theaters* | `Theater, Theatre, theater, thtre, theatrical` |
| $w_2$ | *casinos / gambling* | `Casino, Gambling, Poker, Slots, Betting` |
| $w_3$ | *restaurants / menus* | `eateries, restaurants, menus, diners, seafood, gourmet` |
| **(5) "Developing" (Narrow Gaussian)** | | |
| $\mu$ | *developing* | `methodologies, rapidly, methodology, scalable, accordingly` |
| $w_1$ | *developing symptoms* | `symptoms, leukemia, jaundice, cancer, lymphoma` |
| $w_2$ | *developing countries* | `countries, nations, continent, hemisphere, Nations` |
| $w_3$ | *drug development* | `prophylactic, adjuvant, assays, analgesic, sterilized` |

## G. Prompts and Annotation Instructions

---
**Centroid Description Prompt**

```
You are a meticulous AI researcher conducting an important investigation into patterns
found in language.
You will be given examples where certain tokens or sequences are highlighted between
delimiters like << >>.  These highlighted segments represent the most strongly
activating tokens or spans for a concept.  They may or may not be meaningful on their
own, and sometimes the surrounding context is what carries the real pattern.
Guidelines:
- Analyze the examples and produce one concise natural-language interpretation that
captures the shared latent patterns present in the highlighted text and examples.
- Focus on describing the semantic, syntactic, stylistic, or conceptual pattern uniting
the examples.
- Each example line includes a normalized_activation score between 0 and 1 (and
sometimes relative_to_max), derived from an energy measure.  Higher values indicate that
this example is more important for the concept.
- If the examples are uninformative, do not dwell on them or list them; instead, give
the most concise possible summary of the pattern you can infer across the examples.
- Do not repeat or reference the marker tokens (<< >>) in your interpretation.
- Do not list multiple possible interpretations or speculate; provide a single, clear,
crisp description.
- Keep your interpretation short, direct, and precise.
- Be as specific as possible so that the interpretation of the description is
unambiguous.
RESPONSE FORMAT (STRICT):
- You may optionally include a brief explanation first.
- The FINAL line of your response MUST be exactly of the form:  [interpretation]:  <your
one-sentence description>
- That line:  * MUST start with "[interpretation]:" * MUST be plain text (no markdown
bullets, no code fences, no quotes around it).  * MUST appear exactly once.  * MUST be
the last line of your response.  Do NOT output anything after that line.
Example of a valid final line:  [interpretation]:  A concept describing X that appears
mostly in Y contexts.
Now, follow the instructions carefully adhering to the format outlined above.
The examples:
{examples}
```
---

---

Semantic Neighboring Components Prompt

**System Message**

```
You are evaluating whether nearby latent components in an MFA represent a similar
    underlying concept/theme.

You will be given:
- one MAIN component
- three NEIGHBOR components
Each component is represented by several top examples.

Your task:
1. Determine whether each neighbor shares a semantically similar main theme/concept
    as the MAIN component.
2. Count how many of the 3 neighbors share the theme.

Guidelines:
- Focus on the dominant semantic/theme or dominant structural pattern in the examples.
- A neighbor counts as "shared" if the main theme is clearly related, even if a bit
    different.
- Do not require exact same concept.
- If the main component has no clear theme, then shared_count should usually be 0.
- Ignore 1-2 noisy examples if the majority is clear.
- Be conservative: only count a neighbor if it is clear that they are related.
- The goal is to determine whether regions spanned by multiple gaussians in the MFA
    represent a broader semantic theme or multiple unrelated themes.

Return format (STRICT):
You may include a short explanation first.
The FINAL TWO lines of your response MUST be exactly:
[shared_count]: <0|1|2|3>
[shared_neighbors]: <comma-separated neighbor component ids OR "none">

Rules:
- [shared_count]: must appear exactly once and be the second-to-last line.
- [shared_neighbors]: must appear exactly once and be the last line.
- If shared_count is 0, [shared_neighbors] must be "none".
```

**User Prompt**

```
MAIN COMPONENT: <MAIN_COMPONENT_ID>
<MAIN_COMPONENT_TOP_EXAMPLES>

NEIGHBOR COMPONENT: <NEIGHBOR_COMPONENT_ID_1>
<NEIGHBOR_COMPONENT_TOP_EXAMPLES_1>

NEIGHBOR COMPONENT: <NEIGHBOR_COMPONENT_ID_2>
<NEIGHBOR_COMPONENT_TOP_EXAMPLES_2>

NEIGHBOR COMPONENT: <NEIGHBOR_COMPONENT_ID_3>
<NEIGHBOR_COMPONENT_TOP_EXAMPLES_3>

How many of the 3 neighbors share the same dominant theme/concept as the MAIN
    component?
```

**SAE Feature Relevance Prompt**

**System Message**

```
You analyze neural network interpretability features. For each feature, explain why
    it is or isn't relevant to the given token. Be concise but specific in your
    reasoning.
```

**User Prompt**

```
Analyze which SAE (Sparse Autoencoder) features are semantically relevant to this
    specific token.

TOKEN: "<TOKEN>"
ARTICLE: <ARTICLE>
CONTEXT: "<... left context ...>[<TOKEN>]<... right context ...>"

Active SAE features:
  [<FEATURE_ID_1>]: <FEATURE_DESC_1>
  [<FEATURE_ID_2>]: <FEATURE_DESC_2>
  ...
  [<FEATURE_ID_N>]: <FEATURE_DESC_N>

For EACH feature above, determine if it is RELEVANT (true) or IRRELEVANT (false) to
    the token.

Your response must be a JSON object with this structure:
{
  "features": [
    {
      "atom_id": <integer – the feature ID from the list above>,
      "reasoning": <string – your explanation of why this feature is or isn't
    relevant>,
      "relevant": <boolean – true if relevant, false if irrelevant>
    },
    ... (one entry for each feature listed above)
  ]
}
```

## Annotating Task

In this task, you are given **two groups** of English examples each with 12 examples: **Group A** (which we label as the positive group) and **Group B** (labeled the negative group). Each example includes:

- a **token** (a short piece of text, often a word or a subword such as "App" in "Apple")
- a **context snippet** (the surrounding text where that token appeared), the token will be highlighted with <<token>>
- a **value** (positive or negative).

The examples shown are meant to be the **extremes**: Group A contains examples with the **largest positive values**, and Group B contains examples with the **most negative values**. Your goal is to identify the main theme or pattern in **each group** (if one exists), and then label it with a category from the ones provided below.

**Important:** Occasionally, a few examples may have a value that is **small compared to the maximum magnitude** in that group, or may be of opposite sign (negative vs. positive). In those cases, treat them as noise and **do not include them** when formulating a pattern.

## Pattern types

Patterns can be:

- **Semantic (meaning-based):** The shared pattern depends on **what the text is saying**. If you paraphrase the sentence but keep the meaning, the pattern still holds (same topic, entity type, situation, or expressed idea).
  *Examples:* "countries/cities", "sports teams", "medical terms", "dates/years"
- **Syntactic / stylistic (surface-form-based):** The shared pattern depends on **how the text is written or where the token appears**, and would still hold even if the topic changed completely. It's about format, punctuation, casing, position, or a structural template.
  *Examples:* opening quotes, sentence-initial token, parentheses/citations, capitalization patterns, sentences with the same template
- **None** - no pattern

Notes:
- **A good test is: Imagine paraphrasing the sentence (order, words etc), will it still belong in the set of examples? If yes, its semantic, if not then syntactic.**
- A pattern should be considered "real" only if it appears in the **majority** of the high-magnitude examples in that group. (If you notice 2–3 examples that don't fit, that's okay.)

## To complete the task, please do the following:

### 1) For **Group A (positive)**

Write a short description of the main theme/pattern you see **among the strongest positive examples**.

- If the shared pattern is mostly about the **token itself**, say so.
- If it's mostly about the **context/style/structure**, say so.
- If it's mostly about a **topic**, describe the topic.

Then label Group A as one of:

- **Semantic**
- **Syntactic**
- **None** (you cannot find a meaningful shared pattern)

### 2) For **Group B (negative)**

Do the same for the **strongest negative examples**:

- **Description**
- **Label:** Semantic / Syntactic / None

**Notes:**
- Some words may be uncommon or unfamiliar. If needed, do a quick web search to understand what a word/name refers to.
- Always pay attention to the sentence that the token is actually, for example:

*Figure 8.* Annotation instructions provided to graduate NLP students for labeling the loadings as either semantic or syntactic, for the analysis of §4.

# Annotating Task

In this task, you are given a set of 25 English texts (examples). Each example includes a **token** (a short piece of text often a word) and a **context snippet** (the surrounding text where that token appeared). Your goal is to identify the main **theme or pattern** shared across the examples.

Patterns can be:

- **Shallow**: the theme is very tight and specific (e.g., the same token or a repeated phrase/structure).
- **Broad**: the theme is clear, but the examples cover **multiple subtypes/categories** within that theme (more variety under one umbrella).
- **None**

A pattern should be considered "real" only if it appears in **the majority of examples**. (If you notice 2 or 3 examples that don't fit, that's ok)

To complete the task, please do the following:

1. **Write a short description** of the main theme/pattern you see in the 25 examples.

    - If the shared pattern is mostly about the **token itself**, say so.
    - If it's mostly about the **context/style/structure**, say so.
    - If it's mostly about a **topic**, describe the topic.

2. **Choose one label** for the set:

- **Shallow**: the theme is very tight and specific (the exact same token or a repeated phrase/structure).
- **Broad**: the theme is clear, but the examples cover **multiple subtypes/categories** within that theme (more variety under one umbrella).
- **No theme**: you cannot find a meaningful shared theme or pattern.

Should always look at the token, if its the same token then its shallow, if its a range of tokens within a theme then its broad. Additioanlly, if all the contexts are the same structure with very slight variation then its shallow too. Please note that some of the words might be uncommon words that you are not familiar with. In such cases, you will need to do a quick search over the Web to understand the meaning of words.

*Figure 9.* Annotation instructions provided to annotators for labeling the Gaussians as either broad or narrow, for the analysis of §4.

