# OpenReview forum: "From Directions to Regions: Decomposing Activations in Language Models via Local Geometry"
_ICML.cc/2026/Conference — ICML 2026 regular_

### Official Review · Reviewer_YpXZ · 2026-03-12

**Soundness:** 3
**Presentation:** 2
**Significance:** 3
**Originality:** 3
**Overall Recommendation:** 4
**Confidence:** 2

**Summary:**

The paper proposes using Mixtures of Factor Analyzers (MFA) to decompose language-model activations into region centroids and within-region low-dimensional variation, as an alternative to methods based on global directions. The main idea is that many concepts may be locally structured, nonlinear, or multi-dimensional, and thus are better modeled as neighborhoods of Gaussian regions with local subspaces rather than isolated directions. The authors train large-scale MFAs on Llama-3.1-8B and Gemma-2-2B activations and evaluate the resulting decomposition through qualitative analyses, comparisons to sparse autoencoders, and downstream localization and steering benchmarks. Overall, the paper argues that local geometry provides a more interpretable and practically useful unit of analysis for concept discovery and control.

**Compliance With Llm Reviewing Policy:**

Affirmed.

**Final Justification:**

The paper is well written and easy to follow, and the central idea is interesting and meaningfully distinct. It addresses an important question in representation decomposition and activation steering.

My main concerns were whether the gains truly came from the mixture structure and whether local factor information beyond centroids meaningfully contributes to steering. The rebuttal addressed these points in a useful way: the added global FA baseline helps show the gains are not simply from a more flexible low-rank model, and the follow-up covariance-based steering ablation provides more direct evidence that the local structure is relevant for control. While this evidence is still not fully complete, it makes the core interpretation substantially more convincing. I therefore support the paper at the weak accept level.

**Key Questions For Authors:**

The baseline suite is generally reasonable. Can the authors add two additional classes of baselines?

1. A global factor-analyzer / PPCA / single low-rank Gaussian model, i.e., a variant without mixture structure and with only a single global low-dimensional subspace. This would help isolate whether the gains actually come from modeling local mixture structure, rather than simply from using a more flexible low-rank model than PCA.

2. A centroid-only steering versus steering that also uses local coordinates / local factor information. The current steering results appear to rely primarily on centroid-based interventions. This leaves an important question open: if the paper’s main claim is that local factor variation is an essential part of the representation, why does the steering mechanism mainly use the centroid? An ablation comparing centroid-only interventions against interventions that also incorporate local factor information would clarify whether the local subspace has real control value, or whether most of the practical benefit is already captured by the centroid alone.

**Limitations:**

Yes.

**Strengths And Weaknesses:**

**Strengths:**

1. This method is grounded in a well-defined probabilistic model. The paper clearly explains how MFA induces a decomposition into regional assignment and local variation, and how this differs from global dictionary-style decompositions.

2. This paper benefits from intuitive figures and a coherent narrative connecting geometric motivation, model definition, qualitative analysis, and downstream evaluations. The distinction between centroids and local offsets is especially well communicated.

3. The work addresses an important question in mechanistic interpretability: what the right unit of analysis should be when concepts are not well captured by single global directions.

4. The use of MFA as the core decomposition model for LM activations appears novel in this context, and the paper makes a distinct conceptual move from “directions” to “regions + local subspaces”.

**Weaknesses:**

1. The main concern is that, though current evidence is promising, it is not yet fully conclusive. Several central interpretability claims depend on LLM- and human-based annotation procedures. While the appendix includes agreement analyses, the sample sizes remain modest, and some conclusions—especially around semantic/syntactic structure and interpretability fraction—still rely on subjective judgments.

2. The comparison to SAE in steering is not entirely clean because the intervention forms differ across methods. The appendix motivates these choices empirically, but it remains somewhat unclear how much of the performance gap should be attributed to the representation versus the chosen intervention rule.

3. The empirical scope can be further broadened by adding more analysis on more model families, more sets of layers.


**Minor Issue:** The figures on pages 21 and 22 of the appendix are quite cluttered, with text overlapping and mixed together, which makes them difficult to read.

---

> ### Author Rebuttal · Authors · 2026-03-30
>
> We thank the reviewer for the thoughtful and constructive feedback. We especially appreciate the reviewer’s recognition that the paper makes a distinct conceptual move from global directions to regions with local subspaces. We address the main concerns and suggestions below.
>
> **W1: LLM and human-based analysis**
>
> We understand the concern. For questions such as feature interpretability or semantic vs. syntactic structure, some form of human or LLM judgment is difficult to avoid. In the paper, we use these analyses only for large-scale characterization, while performance claims rely on quantitative results. To make the characterizations fully transparent as to what the results represent, we included in the appendix the exact instructions given to both annotators and the LLM judge. We appreciate the reviewer highlighting this point, and would be happy to provide any additional analysis to address this concern.
>
> **W2: Comparison in causal steering**
>
> As discussed in the appendix, we evaluated each decomposition with the intervention form that is empirically strongest for that method. We view this as the fairest comparison, since different decompositions capture activation structure differently and therefore support different control mechanisms. For MFA, modeling activation space as local regions naturally enables region-based intervention. We see this as part of the contribution rather than a weakness of the comparison. Together with the appendix ablation, we believe this comparison is justified.
>
> **W3: Empirical Scope**
>
> We believe the current scope provides meaningful evidence for the paper’s research question: whether local geometry is a useful and scalable unit for decomposing activation space. In the paper, we trained 12 large-scale MFAs across two model families and multiple scales, and evaluated them through localization, steering, qualitative analysis, and comparisons to both unsupervised and supervised baselines. Overall, the trends are consistent across these settings, which suggests that the results are not specific to a single model or evaluation setup.
>
> Regarding layers, in the large-scale MFA experiments, we focus on selected early and late layers in order to study representations at different stages of the model. At the same time, our localization benchmark is evaluated across nearly all layers, where MFA performs strongly on the large majority of them. For example, on Llama, MFA outperforms DBM on 30/31 layers for continent, 25/31 for country, and 29/31 for language (RAVEL). Taken together, we think these results provide a reasonable empirical basis for the main claims of the paper.
>
> **Q1: Baselines**
>
> Thanks for the great suggestion. We added the global Factor Analysis (FA) baseline in the small-scale settings (RAVEL and MCQA) where this comparison is feasible. Notably, this comparison does not extend to our full large-scale steering setting, where a single global FA is fundamentally limited. The number of features it can represent is bounded by the activation dimension, which is too restrictive for the scale we used of 100 million activations. In that regime, the mixture is not merely helpful but necessary.
>
> We trained FA with the same rank as the corresponding MFA to isolate the added value of the mixture structure. We found that on RAVEL, a single FA matched MFA having the same score. However, on MCQA it reached only 35.0% accuracy for Gemma and 34.8% for Llama, substantially below MFA with 80.2% and 70.5%. This is consistent with our broader finding that RAVEL requires substantially lower rank than MCQA, and further supports the need for locality in more complex settings.
>
> We thank the reviewer for this valuable suggestion, which we will add to the final version of the manuscript.
>
> **Q2: Centroid-only steering**
> This is less straightforward to evaluate directly in steering, since the local subspace is only identifiable up to rotation, its individual axes do not need to align with monosemantic concepts. However, in localization we do provide an ablation showing that removing the subspace causes a large drop on MCQA, from 80% to 39%, which supports the importance of the local subspace. This suggests that the subspace captures meaningful structure that is not present in the absolute position of the region itself (centroid). In addition, Tables 1 & 5 show many cases where individual axes align with sub-concepts within a region, providing further evidence that the subspace contains useful semantic structure that is useful for steering. We believe making use of this structure for control is an interesting direction for future work (and are already seeing significant results in our current research).
>
> **Figures in Appendix**
>
> Thank you for the feedback\! We will recreate the figures so that they are more organized and readable in the revision.
>
> We thank the reviewer again for the careful reading and constructive suggestions. We would be happy to engage regarding any remaining points if helpful.

---

> > ### Author Rebuttal · Reviewer_YpXZ · 2026-04-03
> >
> > Thank you for the detailed rebuttal. I appreciate that the authors have addressed some of my earlier concerns, particularly by adding the global FA baseline and clarifying the broader empirical scope. The new FA results are helpful and largely address my question about whether the gains come from mixture structure rather than simply a more flexible low-rank model.
> >
> > However, my concern about centroid-only steering versus the role of local factor information in steering is not fully resolved. The rebuttal provides indirect evidence that the local subspace is meaningful, but it does not directly demonstrate its control value in steering. A direct steering ablation would still be very helpful. Therefore, I would like to keep my score.

---

> > > ### Author Response · Authors · 2026-04-06
> > >
> > > We are glad that the reviewer’s previous concern regarding whether the gains arise from the mixture structure was addressed. **To address the remaining question, we performed an additional ablation testing whether the covariance structure itself also provides useful steering directions.**
> > >
> > > For a Gaussian with covariance ($WW^{T} + \Psi$), we apply SVD to (W), yielding paired read and write directions (V and U in SVD). We use the read directions for interpretation and the write directions for additive steering. We ran this on MFA-8K for Gemma-2 2B and Llama-3.1 8B at layers 18 and 22, respectively, using 75 randomly sampled Gaussians per model. For each Gaussian, we selected the three axes with the top eigenvalues and evaluated steering with the corresponding write vectors.  Due to rebuttal time constraints, we ran a smaller grid search on the parameters rather than the full analysis of the baselines (SAE, Centroids, DiffMeans).
> > >
> > > The results are included at
> > >
> > > [https://anonymous.4open.science/r/icml\_2026-2C53/steering\_with\_covariance.pdf](https://anonymous.4open.science/r/icml_2026-2C53/steering_with_covariance.pdf)
> > >
> > > where this baseline is denoted **MFA-RW**. We find that **steering with these write vectors produces clear semantic effects and performs comparably to SAE and DiffMeans,** though still weaker than centroid-based steering. We believe this is due both to the smaller grid search and to the fact that these local directions are most effective when applied within the region defined by the Gaussian. We observed similar behavior when exploring this direction internally during the research, but originally left it out because it raises several interesting questions that felt better suited for future work. In particular, directions within a Gaussian often appear to promote specific slices of the broader concept captured by the centroid, and applying these directions in the appropriate local region produces a stronger causal effect than applying them arbitrarily. In many cases, the centroid and covariance directions were nearly orthogonal, which hints at a compositional effect rather than additive.
> > >
> > > We hope these results, together with the localization ablation and the existing qualitative evidence (tables 1 and 5), further support that the learned local structure is meaningful for steering and model control.

---

### Official Review · Reviewer_EZUE · 2026-03-12

**Soundness:** 3
**Presentation:** 3
**Significance:** 3
**Originality:** 4
**Overall Recommendation:** 4
**Confidence:** 3

**Summary:**

This work leverages Mixture of Factor Analyzers (MFA) to model the activation space of large language models, based on the hypothesis that activations are better characterized as a collection of local regions with low-dimensional variation rather than a global set of linear directions. Specifically, each Gaussian component represents a semantic region defined by a centroid and a low-rank subspace that captures the dominant modes of variation within that region. The authors train MFAs on activations from Llama-3.1-8B and Gemma-2-2B to validate this hypothesis and demonstrate that the resulting decomposition yields more interpretable structures in the activation space. Empirically, the MFA-based representation outperforms prior unsupervised baselines such as PCA and SAE, and achieves performance close to supervised approaches. On causal localization and steering tasks, MFA-based features provide strong performance, which proves that modeling activation space through local geometric view can be used as an effective tool for interpretability and LLM steering.

**Compliance With Llm Reviewing Policy:**

Affirmed.

**Final Justification:**

The authors solve most of my concerns especially for my concerns of layer-wise generalization. Therefore, I will change my current score to 4 as the final social.

**Key Questions For Authors:**

(1) The paper evaluates different numbers of Gaussian components, but the factor dimension R is fixed to 10. Could the authors elaborate how sensitive the results are to the choice of R?

(2) For figure 7, you said “Across all settings scores are consistent for all methods, with MFA showing a slight decline.” But I cannot observe this in the figure. Could the authors help me understand this figure?

(3) Could the authors analyze how frequently a single semantic concept is captured by multiple neighboring components rather than a single Gaussian? Are there any quantitive statistics to support this claim?

(4) The distinction between “broad” and “narrow” regions is an interesting observation. Could the authors share some insights to the reason of this distinction? If we prune some weights, will the regions be destroyed?

(5) It would be interesting to understand what kinds of applications this representation decomposition can support in a post-training or training-free setting. For example, can we use the proposed region-based representation enhance specific capabilities such as math reasoning ability of LLMs?

**Limitations:**

As I mentioned in the weaknesses, the empirical validation is relatively limited in scope. Several qualitative analyses rely on small sampled subsets, and experiments are conducted on only two models while primarily analyzing residual stream activations. In addition, both the structural analysis and steering experiments focus on a small number of selected layers, leaving it unclear whether the proposed region-based decomposition and steering behavior generalize across models, activation types, and layers.

**Strengths And Weaknesses:**

**Strengths:**

(1) One of the most compelling one is that, unlike the previous works, this one models LLM activation spaces as a collection of local regions with low-dimensional variation rather than a global set of linear directions. The work also empirically proves that from this perspective, we do achieve better interpretable components. Therefore, I think this work could potentially inspire the future work about activation decomposition and steering.

(2) The paper provides a solid motivation for adopting MFA to model activation decomposition. I am convinced that LLM activations exhibit locally structured variation that cannot be well captured by global linear directions. Moreover, the clarification in Section 5 and empirical comparison with both unsupervised and supervised baseline, the experiments on causal localization and steering tasks further support the practical usefulness of the proposed decomposition.

**Weaknesses:**

(1) Several qualitative analysis in the paper are based on relatively small samples. For example, the semantic analysis of Gaussian components is performed on only 50 sampled components out of the 600 learned components, and each component is characterized using only 25 contexts. While these examples provide useful intuition, it is unclear whether the observations generalize across the full set of learned components.

(2) The experiments focus on only two models including Llama-3.1-8B and Gemma-2-2B and analyze primarily residual stream activations. It remains unclear whether such local geometric view for activation decomposition can be generalized to different model families and scales. Similarly, other types of activations within LLMs like attention outputs, attention head representations are also necessarily to analyzed to show the robustness of the MFA modeling.

(3) We know that representations in different layers of LLMs often exhibit substantially different properties. The paper mainly analyzes residual stream activations from specific layers, and it remains unclear whether the proposed region-based structure holds consistently across layers.

(4) The steering experiments are conducted on only two selected layers (e.g., layer 6 and 18 for Gemma-2-2B) without any motivations. As I mentioned, representations in different layers of LLMs are known to exhibit substantially different properties, it would be helpful to understand whether the proposed region-based decomposition enables effective steering across layers, or whether its effectiveness is limited to specific layers.

---

> ### Author Rebuttal · Authors · 2026-03-30
>
> We thank the reviewer for the careful reading and thoughtful feedback. We appreciate the recognition that the paper offers a compelling local geometric view of activation decomposition and may inspire future work. We address the points below.
>
> **W1: Qualitative analysis size**
>
> To strengthen the qualitative evaluation, we re-ran the analysis with a substantially larger sample, increasing from 50 to 250 components. The resulting trends were similar to those reported in the paper, and the conclusions remained unchanged. We’ll update the manuscript accordingly. Updated figures are available:
>
> [https://anonymous.4open.science/r/icml\_2026-2C53](https://anonymous.4open.science/r/icml_2026-2C53/steering_figure_final_scores.pdf)
>
> **W2: Generalization across models, activation types**
>
> The paper indeed focuses on two representative model families and residual-stream activations. At the same time, we believe the current scope already provides meaningful evidence for the paper’s research question: whether local geometry is a useful and scalable unit for decomposing activation space. We trained 12 large-scale MFAs across two model families and multiple scales, and evaluated them through localization, steering, qualitative analysis, and comparisons to both unsupervised and supervised baselines. The consistency of the overall trends across these settings suggests that the findings are not tied to a single model or evaluation setup. The residual stream has been a central research topic in interpretability in itself. Extending our study to other activation types is very interesting, but proper evaluation would require additional analysis, which would broaden the scope and is thus better suited to future work.
>
> **W3 \+ W4: Generalization across layers**
>
> For large scale training we selected layers at one-third and two-thirds depth. That said, the localization benchmark was evaluated for all layers, where MFA performed strongly on most. For example, on Llama, for RAVEL, MFA outperforms DBM (supervised) on 30/31 layers for continent, 25/31 for country, and 29/31 for language. We didn’t include the full layerwise localization results as MIB reports each task using the maximum score over layers, and we followed that convention. We agree the layerwise picture is informative, and would be happy to include it. If the reviewer is interested in specific full-layer evaluations or analyses, we would be happy to share them.
>
> **Q1: Sensitivity to rank R**
> Based on internal qualitative and quantitative analyses, the results do not appear highly sensitive to R. Increasing R mainly uncovers additional features while largely preserving the main ones already found. This is consistent with the localization results, where performance improves with rank up to a point and then plateaus. Since we train large-scale MFAs, we fixed R=10 as a practical tradeoff between expressivity and computational cost. We’ll include discussion regarding this point in the final version.
>
> **Q2: Figure 7**
> Thank you for pointing this out. Most fluency scores are equal to 1, so variation is minimal and the box plot is hard to read. We’ll revise the appendix and replace the figure with a table, which will make the pattern clearer.
>
> **Q3: Semantic Neighboring Gaussians**
> We followed the reviewer’s suggestion and conducted the analysis. Across our trained 8K MFAs, we tested 670 Gaussians and examined their 3 nearest neighbors for semantic similarity using an LLM judge. We found that 61% had at least one same-theme neighbor, 36% had at least two, and 20% had all three, suggesting that concepts often span multiple nearby Gaussians. We’ll include this analysis in the final version. Please note, Appendix F provides concrete examples sampled from various MFAs that illustrate this, complementing the quantitative results.
>
> **Q4: Broad vs. narrow regions / reason for the distinction**
> We use this distinction because it reflects a recurring difference in the local structure learned by MFA: some Gaussians cover semantically related activations across a broader region of activation space, while others capture higher-density regions. The broad/narrow terminology is intended as a descriptive characterization of this difference. Regarding weight-pruning, we would appreciate further clarification on the setup the reviewer had in mind.
>
> **Q5: Post-training or training-free applications**
> Thanks for the interesting comment. We agree MFA is a promising direction for post-training and training-free applications. Especially for mathematical reasoning, where recent work suggests that arithmetics rely on nonlinear structure that MFA may capture better than existing methods. We believe that alignment and safety are also interesting research areas that could benefit from MFA.
>
> We hope the points address the concerns, particularly regarding the scope of the evaluation and generalization to additional layers. We would be happy to engage regarding any remaining points if helpful.

---

> > ### Author Rebuttal · Reviewer_EZUE · 2026-04-03
> >
> > Thank you for the detailed rebuttal.
> >
> > I still find the layer-wise generalization insufficiently addressed. Key analyses such as steering are conducted on only a few selected layers, while representations in LLMs are known to vary significantly across depth. Since MIB reports the maximum over layers, it may obscure layer-wise variability. Providing full layer-wise results would be important to assess whether the method consistently works across depth.
> >
> > This question is important for understanding the method, I will maintain my current score.

---

> > > ### Author Response · Authors · 2026-04-05
> > >
> > > We are glad that our response resolved most of the reviewer’s concerns. **To address the remaining issue,** **we have repeated our main experiments for all/more layers**.
> > >
> > > **For localization**, we now include the full layer-wise results for both tasks at:
> > >
> > > [https://anonymous.4open.science/r/icml\_2026-2C53/localization\_layerwise.pdf](https://anonymous.4open.science/r/icml_2026-2C53/localization_layerwise.pdf)
> > >
> > > **These results echo the same trend observed in the paper**: MFA consistently outperforms the supervised baseline across almost all layers on all RAVEL tasks and for MCQA it remains competitive with the unsupervised baselines for the task. Overall, this provides strong evidence that MFA successfully recovers features at all layers.
> > >
> > > **For steering**, the large-scale MFAs evaluated in the paper require substantial resources; training MFAs **for all layers** requires storing over 2 billion activations on disk per model (dozens of TBs) and using multiple H100 GPUs for \>100 days. This is similar to training large-scale SAEs \[1, 2\], which were practically done by large labs and companies and are infeasible within our academic budget. However, to further demonstrate the robustness of MFA, **we trained smaller-scale MFAs (250 Gaussians) across 12 additional layers (6 per model), covering early, middle, and upper layers**. We then evaluated them on steering using the same setup from the paper and compared them against state-of-the-art SAEs \[1,2\]. We include these results at:
> > >
> > > [https://anonymous.4open.science/r/icml\_2026-2C53/layerwise\_steering.pdf](https://anonymous.4open.science/r/icml_2026-2C53/layerwise_steering.pdf)
> > >
> > > **These additional results show the same trend reported in the paper, namely, MFA steering performance is consistently stronger than that of SAEs across all layers.**
> > >
> > > We hope these additional results resolve the reviewer’s concern, as they show that **MFA’s localization and steering results are not limited to only a specific set of layers**.
> > >
> > > \[1\] Gemma Scope: Open Sparse Autoencoders Everywhere All At Once on Gemma 2
> > > \[2\] Llama Scope: Extracting Millions of Features from Llama-3.1-8B with Sparse Autoencoders

---

### Official Review · Reviewer_6W9i · 2026-03-13

**Soundness:** 3
**Presentation:** 4
**Significance:** 4
**Originality:** 3
**Overall Recommendation:** 5
**Confidence:** 4

**Summary:**

This paper proposes using Mixture of Factor Analyzers (MFA) as an unsupervised interpretability method for decomposing and steering language model activations. Rather than searching for global directions (as in sparse autoencoders or PCA), MFA models activations as grouped in local clusters. The method partitions the activation space into local Gaussian regions, each equipped with a low-rank subspace capturing within-region variation. The method is evaluated on Llama-3.1-8B and Gemma-2-2B for localization and steering benchmarks, where it outperforms SAE baselines and is even competitive with supervised methods.

**Compliance With Llm Reviewing Policy:**

Affirmed.

**Final Justification:**

I believe this paper should be accepted. The new results provided during the rebuttal did not lead to a substantial change in my perception of the paper regarding impact and novelty, so I did not raise my score from 5 to 6.

**Key Questions For Authors:**

- Is the notion of broad and narrow related with the actual variance of the Gaussian components? Did you run experiments to verify this?
- Can you provide more intuition why MFA works better than SAEs and other methods? Do you think it fundamentally matches the structure of activations more closely or it simply offers more expressivity to capture the complexity of these representations?
- Did you explore the impact of rank on localization and steering performance? To what extent should the rank be tuned?

**Limitations:**

Yes.

**Strengths And Weaknesses:**

Overall this paper consists of a well-executed and elegant idea that delivers strong results on the localization task and seems promising on the steering setup. It also offers a new perspective on the structure of model activations, enabling a two-level interpretability scope and providing insights on modle analysis.

### Strengths
- Clear and well-written exposition. The MFA framework is introduced and motivated carefully. The connection between the statistical model and the interpretability goals is intuitive, and the decomposition into centroid (region) and loading (local variation) components is well-illustrated and sensible.
- Strong empirical results on localization. MFA outperforms PCA and SAEs by substantial margins on RAVEL and is competitive with DAS, a supervised method, which is a meaningful result for an unsupervised approach.
- Interpretable decomposition. The interpretability fraction metric (IF = 0.96 for MFA vs. 0.29 for SAEs) is a compelling finding, and the centroid/loading examples in Tables 1, 5, and 6 provide convincing qualitative support.

### Weaknesses
- Sensitivity to architectural choices is unexplored. Both evaluated models (Llama-3.1-8B and Gemma-2-2B) differ in several ways beyond scale, including their use of pre-norm vs. post-norm and attention variants. It is unclear whether the differences observed between the two models reflect something fundamental about the models' representations or are artifacts of how the method reacts to architectural choices. Even a brief discussion of this would help situate the findings.
- Steering results show low scores and high variance (Figure 5). The absolute scores in Figure 5 are low across all methods, and the variance is large enough that many comparisons are difficult to interpret confidently. While MFA does appear to outperform SAEs and DiffMeans in most settings, the reliability of this conclusion is weakened by the variance. A more refined experimental setup, or at minimum an analysis of what drives the variance, would make these results more convincing.

---

> ### Author Rebuttal · Authors · 2026-03-30
>
> We thank the reviewer for the careful reading and thoughtful feedback. We appreciate the reviewer’s positive assessment of the paper as a well-executed and elegant approach, the recognition of its strong empirical results, and the view that it offers a useful new perspective on activation structure.
>
> **W1: Sensitivity to architectural choices**
>
> Our goal in the current version was to test whether locality serves as a strong inductive bias for decomposing activations across two fairly different model families, rather than to isolate the effect of specific architectural choices. Considering the differences observed, mainly in the proportions of broad and narrow Gaussians, we do find it a particularly interesting direction for future work. In particular, Gemma-2-2B and Llama-3.1-8B differ not only in scale but also in architectural choices such as normalization and attention design. Prior work \[1, 2\] suggests that normalization affects the semantic organization and geometry of representations. It therefore may be that the observed differences reflect either underlying representational differences between the models or differences in how locality manifests under different architectural constraints. We agree with the reviewer that it is a particularly interesting direction for future work and will incorporate a discussion regarding this point in the final revision.
>
> **W2: Steering results have low absolute scores and high variance**
> We aimed to stay as close as possible to the original evaluation setup in AxBench [3], so that our evaluation remains well aligned with prior work. The relatively high variance is consistent with what is also observed in the original AxBench paper. One likely reason is that the metric uses a coarse, discrete 0–2 scale, so even small random differences between generated outputs can lead to different assigned scores. This can make the measurement somewhat noisy under stochastic generation.  We nevertheless followed this protocol closely in order to keep the evaluation consistent with existing literature.
>
> **Q1: Broad vs. narrow regions and actual variance of Gaussian components**
> We appreciate this question. Our original distinction between broad and narrow was primarily based on semantic characterization rather than a rigorous quantitative definition. Motivated by the reviewer’s suggestion, we examined whether these categories correlate with component variance, but found no such correlation. We believe this is a result of the low rank used in the Gaussians (r=10), which is often lower than the region’s intrinsic rank. Thus, the total variance captured by a component is constrained by the rank limit and may not fully reflect how broadly or narrowly the component is distributed in activation space.
>
> **Q2: Why MFA works better than SAE and related methods**
> Our current intuition is that by modeling the activation density directly, MFA provides causal handles that better respect the model’s natural activation distribution. This makes it possible to steer more strongly without pushing activations out of distribution, and therefore without sacrificing fluency. This is also consistent with the results in the appendix, where MFA achieves substantially higher causal concept scores while maintaining fluency comparable to the other methods.
>
> Additionally, we repeatedly observed that the same concept can appear as different directions, sometimes even nearly orthogonal ones, in different regions of the activation space. This suggests that locality may be an important inductive bias for capturing how concepts are naturally represented by the model.
>
> **Q3: Impact of rank R**
> This is an interesting question. We find the localization results to be most insightful in this regard. For RAVEL, as the ablation shows, the centroids captured the concepts best, so the rank did not need to be large; r \= 10 was sufficient to model the local activation density and achieve strong results. However, for MCQA, we found that a substantially larger rank (r=50) was needed. Our intuition is that MCQA involves a pointer-like feature that corresponds to a lower-variance direction in activation space, so representing it with MFA requires a higher rank.
>
> Overall, increasing the rank mainly seems to add additional directions of structured variation present within each region. Therefore, the appropriate rank depends on the intended use of MFA, whether for modeling the activations’ geometry, localization, reconstruction, or steering.
>
> We thank the reviewer again for the constructive suggestions and engaging discussion. We would be happy to discuss any remaining points or answer any additional questions.
>
> \[1\] Transformer Normalisation Layers and the Independence of Semantic Subspaces
>
> \[2\] Geometric Interpretation of Layer Normalization and a Comparative Analysis with RMSNorm
>
> \[3\] Geometric Interpretation of Layer Normalization and a Comparative Analysis with RMSNorm

---

> > ### Author Rebuttal · Reviewer_6W9i · 2026-03-31
> >
> > Thank you for the clarifications. The rebuttal is satisfying, but it did not lead to noticeable changes in the paper / new results that would improve the quality of the paper in my opinion. My score remains unchanged.

---

> > > ### Author Response · Authors · 2026-04-06
> > >
> > > We are glad that the rebuttal was satisfying, and thank the reviewer for the engaging discussion.

---

### Official Review · Reviewer_1sJk · 2026-03-13

**Soundness:** 3
**Presentation:** 3
**Significance:** 3
**Originality:** 3
**Overall Recommendation:** 5
**Confidence:** 4

**Summary:**

This work proposes using Mixtures of Factor Analyzers (MFA) to disentangle activations into interpretable low-dimensional Gaussian subspaces and their local axes of variation, thus introducing a new approach relying on local geometry rather than a dictionary of isolated directions. Authors provide extensive evaluation of the proposed method on LLaMa-3.1-8B and Gemma-2-2B, showing superior performance over other unsupervised methods at localization and causal steering tasks. For better clarification of how MFA’s decomposition differs from dictionary learning, authors conduct a thorough comparison with SAEs in analysis section.

**Compliance With Llm Reviewing Policy:**

Affirmed.

**Final Justification:**

My main concerns were addressed.

**Key Questions For Authors:**

Can you please address the missing baselines weakness, so i can raise the score.

**Limitations:**

yes

**Strengths And Weaknesses:**

### Soundness
- Submission is technically sound and claims are well supported by the extensive experiments including localization and causal steering benchmarks, analysis of activation structures discovered by MFA and side-by-side comparison with SAE features.
- Several baseline methods are missing from the causal steering evaluation and related work discussion: ReFT-r1 from AxBench paper and simple prompting baseline.

### Presentation
Paper's narrative is easy to follow with proper positioning in the context of prior works and additional experimental details are listed in the Appendix.

### Significance
Work addresses a very important problem of models's mechanistic interpretability. It introduces a new approach for practical activation decomposition in LMs, relying on local geometry rather than a dictionary of isolated directions. Moreover, it provides a thorough analysis of which type of activation structures are learnt by MFA and how they are different from usual dictionary learning, which allows for further research in this area.

### Originality
Work provides novel insights to understanding how LLM's activations can be better disentangled and uses existing in literature techniques to evaluate the proposed method.

---

> ### Author Rebuttal · Authors · 2026-03-30
>
> We thank the reviewer for the careful reading and helpful feedback. We appreciate the positive assessment of the paper’s framing and empirical results, as well as the recognition of its broad evaluation. We also thank the reviewer for pointing out additional baselines that would further strengthen the paper.
>
> **Missing baselines in causal steering evaluation (ReFT-r1 and prompting baseline)**
> Thank you for this suggestion. We agree that these are important baselines to include to better position the method’s steering. Following the reviewer’s suggestion, we have evaluated both baselines in the same settings and will include the results in the manuscript. For the prompting baseline, in order to steer a completion model, we appended to each prompt a prefix: *“In the following sentences I will discuss the concept: \<concept\>.*” to the base prompt used in the paper “I think that” and evaluated on the generated content without the prefix, identically to the setting in the paper. For ReFT, we used the provided 500 concepts and respective training data from Axbench and trained for the models and layers used in the steering evaluation in the paper. We found that both ReFT and prompting outperform all unsupervised methods, consistent with the results of Axbench. We attached a link to the updated figure: [https://anonymous.4open.science/r/icml\_2026-2C53](https://anonymous.4open.science/r/icml_2026-2C53/steering_figure_final_scores.pdf)
>
> We will include these results in the final revision and additional discussion in Related Work to better situate the steering experiments within prior work.
>
>
> We thank the reviewer again for the constructive feedback and hope that including the baselines and expanding the discussion addresses the main concern.

---

> > ### Author Rebuttal · Reviewer_1sJk · 2026-04-04
> >
> > Thank you, i will raise my score

---

> > > ### Author Response · Authors · 2026-04-06
> > >
> > > We are glad the additional baselines addressed the concern, and thank the reviewer for the constructive feedback.

---

### Decision · Program_Chairs · 2026-04-30

**Decision:**

Accept (regular)

**Comment:**

All four reviewers support acceptance, and post-rebuttal discussion confirmed that remaining concerns were adequately addressed. The paper introduces a conceptually distinct approach to activation decomposition, moving from global directions to local regions with low-rank subspaces via MFA. It delivers strong empirical results, outperforming unsupervised baselines and competing with supervised methods on localization while achieving stronger steering than SAEs. The analysis connecting local geometry to interpretability is thorough and well-motivated. The authors were responsive throughout, providing additional layer-wise results, baselines, and ablations that strengthened the paper. I recommend acceptance.